# Partial energy balance closure of eddy covariance evaporation measurements using concurrent lysimeter observations over grassland

Peter Widmoser[1], Dominik Michel[2]

[1]Institute of Natural Resources Conservation, Department of Hydrology and Water Resources Management, Kiel University, 24118 Kiel, Germany
[2]Institute for Atmospheric and Climate Science, ETH Zurich, 8092 Zurich, Switzerland

*Correspondence to*: Dominik Michel (dominik.michel@env.ethz.ch)

**Abstract.** With respect to the ongoing discussion on the causes of the energy imbalance and approaches to force energy balance closure a method had been proposed which allows the partial latent heat flux closure (Widmoser and Wohlfahrt; 2018). In the present paper, this method is applied to four measurement stations over grassland under humid and semi-arid climate, where lysimeters ($LY$) and eddy covariance ($EC$) measurements were taken simultaneously.

Results differ essentially from the ones quoted in literature. We distinguish between resulting $EC$ values being weakly and strongly correlated to $LY$ observations as well as systematic and random deviations between $LY$ and $EC$ values. Overall, an excellent match could be achieved between $LY$ and $EC$ measurements, after applying evaporation-linked weights. But there remain large differences between standard deviations of $LY$ and adjusted $EC$ values. For further studies we recommend data collected at time intervals even below half an hour.

No correlation could be found between evaporation weights and weather indices. Only for some datasets a positive correlation between evaporation and the evaporation weight could be found. This effect appears pronounced for cases with high radiation and plant water stress.

Without further knowledge on the causes of energy imbalance one might perform full closure using equally distributed weights. Full closure, however, is not dealt with in this paper.

## 1 Introduction

Non-closure of the surface energy balance, i.e. the sum of latent ($LE$) and sensible ($H$) heat exchange falling short of available energy ($A$), is a common issue in eddy covariance flux *(EC)* measurements. Available energy equals net radiation ($RN$) minus the soil heat flux ($G$) and any other energy storage (Wohlfahrt and Widmoser, 2013). At the majority of eddy covariance flux sites it is the rule rather than the exception to find that the sum of the turbulent fluxes $LE + H$ underestimates $A$ by 20-30 % (Leuning et al., 2012; Wilson et al., 2002). This apparently systematic bias has been extensively discussed in literature (see reviews by Foken, 2008; Foken et al., 2011; Leuning et al., 2012, Mauder et al., 2020). In the last, most recent review, the

following classification of reasons for the energy gap problem is listed: 1) instrument error, 2) data processing error, 3) additional sources of energy, 4) secondary circulation of energy. Own hourly observations show that the bulk of $LE+H$ underestimates is detected around noon, whereas during sunrise and sunset also overestimates are observed.

There are two practical approaches to deal with the energy imbalance problem: 1) to compare $EC$ measurements with

concurrent lysimeter measurements and 2) using models.

Lysimeters ($LY$) have a long tradition in hydrology and micrometeorology and their limitations and sources of uncertainty are well known. There usually is a very strong correlation between concurrent $LY$- and $EC$-based evaporation data, with the $LY$ values generally being higher. An overview of efforts to compare $EC$ evaporation to lysimeter measurements can be found in Gebler et al. (2015). A few of these studies related to this article are quoted below.

Chavez and Howell (2009) hint at various error sources for $LY$ and $EC$ measurements. $EC$ observations on cotton fields in Texas with quarter-hourly measurements resulted in an energy balance gap of 22.0 to 26.8 %. Those gaps were closed assuming Bowen ratio preservation and correct measurements of the available energy. After forced closure of the energy balance, the difference between daytime $LY$ and $EC$ data on two fields could be reduced from -28.8 % to 6.2 %, respectively from -26.0 % to -12.3 %, with an accuracy of $0.03 \pm 0.5$ mm d$^{-1}$ ($\approx 0.9 \pm 14$ Wm$^{-2}$), respectively $-0.1 \pm 0.4$ mm d$^{-1}$ ($\approx -2.8 \pm 11$ Wm$^{-2}$).

Negative values indicate that the lysimeter values were higher on average than $EC$ values.

Evett et al. (2012), using data from the same site as Chavez and Howell (2009), quote errors of daytime $EC$ measurements for latent heat flux of 1.9 to 2.7 mm d$^{-1}$ ($\approx 55$ to 78 Wm$^{-2}$), for sensible heat flux of 1.4 to 1.9 mm d$^{-1}$ ($\approx 40$ to 55 Wm$^{-2}$). They reported substantially larger $LY$ evaporation rates compared to the $EC$ measurements due to differences in plant growth in the $LY$ and the $EC$ footprint. After forced closure of the energy gap as done by Chavez and Howell (2009) mean differences from

-17.4 to -18.7 % were found between the two measurements methods after correcting for plant growth.

In the same way, Ding et al. (2010) closed the energy gaps using half-hourly daytime data on irrigated maize in an arid area in NW-China. There also, differences of daily measurements were reduced by forced Bowen ratio closure of the $EC$ gap. Differences could be reduced from -22.4 % to -6.2 %, the lysimeter measurements again being higher on average.

The following authors dealt with comparing measurements on grassland. Gebler et al. (2015) assumed that the energy balance deficit is caused by an underestimation of the turbulent fluxes only, which are corrected according to the evaporative fraction $LE/(LE+H)$ averaged over 7 days. After correction, they find an agreement of $LY$ with $EC$ values with a total difference of 3.8 % (19 mm) over a year. The best agreements on the basis of monthly values during summer were obtained with less than 8 % of relative errors. The remaining differences are suspected to be due to different plant height within the $EC$ fetch and the

lysimeter. Mauder et al. (2018) evaluated two adjustment methods to close the energy balance: (1) the Bowen ratio preservation adjustment, following the approach of Mauder et al. (2013); (2) the method by Charuchittipan et al. (2014), which attributes a larger portion of the residual to the sensible heat flux. They also compare the $EC$ values with the results of the hydrological

model GEO top 2.0 (Endrizzi et al.; 2014). They found that a daily adjustment factor leads to less scatter than a complete partitioning of the residual for every half-hour time interval.

In the compilation of literature above, the *LY-EC* comparisons relied on the assumptions that the available energy observations are correct and that the Bowen ratio can be preserved. In contrast to the closure method used by the above quoted authors, Widmoser and Wohlfahrt (2018) achieved a partial latent heat closure of the energy balance by combining both, the model and lysimeter-approach, which is afterwards fully closed under the assumption of preservation of the Bowen ratio.

The objective of this article is to extend the above mentioned method, which was applied to one station only, to more stations,
in order to test its applicability and compare its results.

## 2. Material and methods

### 2.1 Measurement stations and data sets

The following Table 1 specifies the stations from which data were used.

**Table 1: Specifications of data used; *SM* denotes soil moisture**

| | | | | | | | | | | | |
|---|---|---|---|---|---|---|---|---|---|---|---|
| **vegatation** | humid grassland | | | | | semi-arid grassland | | | | | |
| **time intervals** | 1h | | 1h | | 1h | 0.5h | | | | | |
| **diurnal obs. time** | 5 am to 8 pm | 9 am to 4 pm | 5 am to 8 pm | | 5 am to 8 pm | 9 am to 4 pm | | | | | |
| **Number of records used** | 1852 | 889 | 720 | 846 | 920 | 1103 | 1126 | 823 | 1186 | 455 | 731 |

| Name of station | Abbreviation | Country | Location | Observation period |
|---|---|---|---|---|
| Graswang | G1 | Germany | 47.57°N, 11.03°E; 864 m a.s.l. | 02.03 – 31.10.2013 |
| | G2 | | | 01.04 – 31.10. 2014 |
| Fendt | F1 | Germany | 47.83°N, 11.06°E 597 m a.s.l. | 01.03 – 24.10. 2013 |
| | F2 | | | 01.04 – 31.10. 2014 |
| Rietholzbach | RHB | Switzerland | 47.37 °N, 8.99 °E, 795m a.s.l | 01.05 – 30.10.2013 |
| Majadas | M1 (dry season) | Spain | 39.56° N, 05.46 W 264 m.a.s.l. | 15.05 – 12.10.2016 |
| | M2 (dry season) | | | 15.05. – 25.08.2017 |
| | M3 (rainy season) | | | 25.08. 2017 – 05.01.2018 |
| | M4 (dry season) | | | 21.04. – 03.09.2018 |
| | M4$_{SM\_moist}$ | | | 21.04. – 03.07.2018 |
| | M4$_{SM\_dry}$ | | | 04.07. – 03.09.2018 |

Data were obtained from the following Institutions: (1) Graswang (*G*) and Fendt (*F*) from M. Mauder, Institute of Technology (KIT-Karlsruhe), Garmisch-Partenkirchen, and R. Kiese, Institute for Technology, Institute of Meteorology and Climate, both Germany; (2) Majadas (*M*) from M. Migliavacca and O. Perez-Priego, Max Planck Institute for Biogeochemistry, Jena, Germany; (3) Rietholzbach (*RHB*) from S. I. Seneviratne and M. Hirschi, Institute for Atmospheric and Climate Science, ETH Zurich.

### 2.1.1 Graswang and Fendt

The stations Graswang and Fendt are both located in grassland ecosystems mostly used for fodder and hay production in the Ammer catchment area in the south of Germany. These sites belong to the Bavarian Alps/pre-Alps Observatory of the TERrestrial Environmental Observatories (TERENO) network (Zacharias et al., 2011), and are part of the Integrated Carbon Observation System (ICOS, icos-infrastruktur.de). The soil in Fendt is classified as cambic Stagnosol, mean annual precipitation and temperature in 2013–2014 were 922 mm and 8.7 °C, respectively. The soil in Graswang is classified as fluvic calcaric Cambisol, mean annual precipitation and temperature in 2013–2014 were 1238 mm and 6.7 °C, respectively. In both cases the site management at the EC tower and on the lysimeters followed the farmers' practices. The practice in Fendt was extensive (two cuts and two manure applications), while it was intensive in Graswang (five cuts and four manure applications, Mauder et al., 2018).

The equipment used in this study is identical for both stations. *EC* instrumentation comprises a CSAT-3 sonic anemometer (Campbell Scientific Inc. USA) and LI-7500 infrared gaz analyzer (LI-COR Biosciences, USA) at 2 m above ground. Available energy (Wm$^{-2}$) was observed using a CNR4 net radiometer (Kipp & Zonen, The Netherlands) at 2 m above ground and the average of three HFP01-SC heat flux plates (Hukseflux, The Netherlands) at a depth of 0.08 m. Spatially averaged soil moisture data (m$^3$m$^{-3}$) were obtained with three CS616 soil moisture sensors (Campbell Scientific Inc. USA) at a depth of 0.06 m. Lysimeter evaporation (Wm$^{-2}$) was obtained with a lower boundary-controlled TERENO-SOILCan large weighing lysimeter (METER Group AG, Germany; described by Gebler et al., 2015 and Mauder et al., 2018), with a surface area of 1.0 m$^2$ and a depth of 1.5 m. The temporal resolution of all data from these stations is one hour.

### 2.1.2 Rietholzbach

The hydrometeorological research station Rietholzbach is located in northeastern Switzerland in a hilly, pre-alpine catchment draining an area of 3.31 km$^2$. The region is characterized by a temperate humid climate with a mean annual precipitation and air temperature of 1438 mm and 7.1°C, respectively, based on the long-term mean 1976-2015. The soil type and depth exhibit a high spatial variability. Overall, shallow Regosols dominate on steep slopes, deeper Cambisols are found in flatter areas, and gley soils are located in the vicinity of small creeks. On the slopes and along creeks, in about 25 % of the area, forest dominates. The remaining catchment area is mostly grassland and partially used as pasture (Hirschi et al., 2017).

*EC* fluxes were measured with a CSAT3 sonic anemometer (Campbell Scientific Inc. USA) and a LI-7500 infrared gaz analyzer (LI-COR Biosciences, USA) at 2 m above ground. Net radiation was measured using two CM21 pyranometers (Kipp & Zonen, The Netherlands) for the net shortwave radiation and two CG4 net radiometers (Kipp & Zonen, The Netherlands) for the net longwave radiation, both at 2 m above ground. The soil heat flux was calculated as the average of three HFP01 and one HFP01-SC heat flux plates (Hukseflux, The Netherlands) at a depth of 0.05 m. The Rietholzbach large weighing lysimeter has a surface area of 3.1 m$^2$ and a depth of 2.5 m including a gravel filter layer at the bottom and gravitational discharge. The

temporal resolution of all data from this station is one hour. For more information on this station refer to Seneviratne et al., 2012 and Hirschi et al. 2017.

### 2.1.3 Majadas

The station Majadas del Tiétar North is located in a Mediterranean tree-grass savannah in western Spain. It is part of the
FLUXNET network (fluxnet.ornl.gov). The vegetation cover is composed of trees (mostly Quercus ilex (L.), approx. 22 trees/ha) and an herbaceous stratum composed by native annual species of the three main functional plant forms (grasses, forbs and legumes). The soil is classified as an Abruptic Luvisol, mean annual precipitation and temperature are 650 mm and 16 °C, respectively (Perez-Priego et al., 2017).

*EC* fluxes are obtained with a Gill R3-50 sonic anemometer (Gill Instruments Ltd., UK) and a LI-7200 infrared gaz analyzer
(LI-COR Biosciences, USA) at 15.5 m above ground. Available energy was observed using a CNR4 net radiometer (Kipp & Zonen, The Netherlands) and the average of four HFP01-SC heat flux plates (Hukseflux, The Netherlands) at a depth of 0.03 m. Spatially averaged soil moisture data were obtained with two Enviroscan soil moisture sensors (Sentek, Australia) at a depth of 0.40 m. Lysimeter evaporation data are the spatial average of four lower boundary-controlled large weighing lysimeters (Umwelt-Geräte-Technik GmbH, Germany) with a surface area of 1.0 m² and a depth of 1.2 m. The used temporal
resolution of all data from this station is one hour (aggregated from half-hourly values). For more information on the station refer to Migliavacca et al. (2017) and Perez-Priego et al. (2017).

Figure 1 and Table 1 give an overview of the locations of the stations and time periods used. Note that for *G1*, *F1, F2* and *RHB* measurements between 5 am and 8 pm were used. The daytimes used for *G2* and Majadas were reduced to 9 am to 4 pm
for reasons given below (Section 2.4). Figure 2 shows the mean daytime course of *A*, *H*, *LY*- and *EC*-based *LE* as well as the resulting energy gap $\varepsilon$ at all four stations.

### 2.2 Possible errors of lysimeter observations

The lysimeters used in this study can achieve measurement accuracies equivalent to between ± 7 and ± 20 Wm⁻², depending on their construction. Furthermore, hydraulic conditions (cylinder walls, soil conditions, ground water table) of the lysimeter
do not correspond with the undisturbed surrounding. In addition to these systematic errors, random errors may occur due to instabilities caused by wind gusts. One may also note that lysimeter observations generally do not include negative values (condensation). The influence of wind and dew on lysimeter observations is described in Meissner et al. (2007) and Ruth et al. (2018). The theoretical accuracy of lysimeter measurements can be calculated from the surface area and weighing accuracy. For the *RHB*-lysimeter (operational since 1976), a systematic accuracy of about 0.03 mm (equivalent to approx. ± 20Wm⁻²
within an hourly interval) is quoted by Hirschi et al. 2017. All other lysimeters of this study (in *F*, *G* and *M*) have a calculated systematic accuracy of 0.01 mm (equivalent to approx. ± 7 Wm⁻² within an hourly interval).

**2.3 Possible errors of *EC* observations.**

Systematic measuring errors of the latent heat flux (*LE*) may be around $\pm$ 30 Wm$^{-2}$, of sensible heat flux (*H*) around $\pm$ 13 Wm$^{-2}$ and of available radiation around $\pm$ 12 Wm$^{-2}$ (Alfieri et al., 2012).

Errors caused by non-closure of the energy balance $\varepsilon = A - LE - H$ are not included in the estimates given above. The $\varepsilon$-errors result as the sum of *A, LE* and *H* errors and may be around $\pm$ 55 Wm$^{-2}$.

**2.4 Data selection**

High quality data were at disposal from all the observation stations. Still we had to dismiss 2 to 5 % of the *EC* measurements - mostly for morning and evening hours with high instability of turbulent fluxes. We sorted them out on the basis of the Out-of-Bound concept introduced by Wohlfahrt and Widmoser (2013), which excludes physically unrealistic measurements. According to this concept, the ratio $r_1 = (r_a + r_c)/r_a$, where $r_a$ and $r_c$ denote aerodynamic and canopy resistance, must numerically be within the range of 1 to infinity (see Fig. 1 in Wohlfahrt and Widmoser, 2013). Case 2 represents $r_1 < 0$ and case 3 represents $0 < r_1 < 1$. Data corresponding to case 2 and 3 are thus omitted. Furthermore, data showing big differences between *LY* and *EC* measurements (i.e. $> 300$ Wm$^{-2} \approx > 0.44$ mm h$^{-1}$) along with strong wind gusts ($> 2.0$ ms$^{-1}$), as well as early morning values with high air humidity and high dew formation were also excluded, thus reducing the original data sets for another 5 % at the average.

The overall data selection led to a reduced number of early morning and late evening data as compared to the number of data available for the rest of the day. That means that results for around sunrise and sunset are generally less reliable. In case of *G2* the morning and evening data had to be reduced to such an extent that we decided to evaluate only data from 9 am to 4 pm. For Majadas, all morning data were omitted for this reason. The numbers of data given in Table 1a correspond to the data analyzed below.

In order to extend the daily time window of analyzed Majadas data (i.e. from 5 am to 8 pm) in the *M4* dataset (dry season), the morning values were corrected for dew-effects. In this way we obtained $w_{LE}$ estimates ($w_{LE\_long}$ ca. 0.4, see Fig. 8b), which compare well with the results of the other stations.

**2.5 Evaluation of weights $w_{LE}$ by regression (partial closure)**

Wohlfahrt and Widmoser (2013) introduced a simple framework for studying the energy imbalance ($\varepsilon$), i.e.

$$\varepsilon = A - H - LE \qquad (1)$$

They proposed three dimensionless weights ($w_A$, $w_H$ and $w_{LE}$) for the terms on the RHS of Eq. (1) which obey the following two constraints: (i) each weight is bound between zero and unity and (ii) the three weights sum up to unity.

Provided these weights are known, the terms on the RHS of Eq. (1) can be corrected for the lack of energy balance closure as:

$$A_c = A - w_A \varepsilon \qquad (2a)$$
$$H_c = H + w_H \varepsilon \qquad (2b)$$
$$LE_c = LE + w_{LE} \varepsilon \qquad (2c)$$

In this paper, we are concerned only with the evaluation of $w_{LE}$ (Eq. 2c) by regressing the difference between $LY$ and $EC$ latent heat fluxes as a function of the energy imbalance:


$$LE_{LY} - LE_{EC} = w_{LE} \varepsilon + d, \qquad (3)$$

where $LE_{LY}$ and $LE_{EC}$ denote the latent heat flux from $LY$ and $EC$ measurements, respectively, $w_{LE}$ represents the slope of the best-fit linear relationship and the y-intercept ($d$) might be interpreted as a systematic difference between $LY$ and $EC$ latent

heat flux measurements. The random difference follows from

$$d_{rand} = LE_{LY} - (LE_{EC} + w_{LE} \varepsilon + d) \qquad (4)$$

For regression, data were binned according to the magnitude of $LE$ in such a way that for each bin the same number of data

pairs ($LY$-$LE$) vs $\varepsilon$, see Eq. (3), was available. The number of bins, i.e. 5 to 14, depended on the number of data per dataset at disposal. At least 90 data-pairs entered each regression.

## 2.6 Used parameters

The results of the partial energy closure will be represented by the following parameters:

-   $D_o = LE_{LY} - LE_{EC\_o}$ as difference between observed $LY$ and observed $LE_{EC\_o}$ values.
-   $D_c = LE_{LY} - LE_{EC\_c}$ as difference between observed $LY$ and corrected $LE_{EC\_c}$ values: $LE_{EC\_c} = LE_{EC\_o} + w_{LE}\varepsilon$.
-   $D_a = LE_{LY} - LE_{EC\_a}$ as difference between observed $LY$ and adjusted $LE_{EC\_a}$ values: $LE_{EC\_a} = LE_{EC\_c} + d$


Furthermore we list the

-   systematic deviations $d$, see intercept in Eq. (3)
-   $\varepsilon_{red}/\varepsilon$ as a measure for the relative $\varepsilon$, remaining after adjustment; $\varepsilon_{red} = \varepsilon\,(1 - w_{LE})$
-   weight $w_{LE}$


One may note that the $D_a$ values correspond to the remaining differences after $LE_{EC}$ adjustment to the $LY$ data and as such may be interpreted as random deviations $d_{rand}$ or noise.

## 3. Results

### 3.1 Basic evaporation characteristics

Tables 2a and 2b give means and standard deviations ($SD$) of the observed $LE_{EC\_o}$, the corrected $LE_{EC\_c}$, the adjusted $LE_{EC\_a}$ and $LY$ evaporations for the analyzed periods and stations along with energy balance deficit $\varepsilon$ and correlation coefficients between $LY$ and $EC$ data. They highlight the substantial difference between the humid and dry stations in terms of the mean magnitude of evaporation. Under moist soil conditions ($M4$), in contrast, the dry station Majadas ranges around the same magnitude as the humid stations.

**Table 2a: Basic evaporation characteristics for the humid stations ($\rho$ = correlation coefficient)**

| | | | G1 | G2 | F1 | F2 | RHB |
|---|---|---|---|---|---|---|---|
| $LE_{EC\_o}$ | [Wm$^{-2}$] | mean | 153.2 | 149.1 | 107.3 | 133.3 | 139.3 |
| | | SD | 99.5 | 78.3 | 95.1 | 73.3 | 100.7 |
| | | $\rho(LE_{LY},LE_{EC\_o})$ | 0.894 | 0.879 | 0.963 | 0.912 | 0.887 |
| $\varepsilon$ | [Wm$^{-2}$] | mean | 64.38 | 100.16 | 59.15 | 87.03 | 25.87 |
| | | SD | 57.81 | 56.78 | 66.52 | 57.75 | 54.50 |
| $LE_{EC\_c}$ | [Wm$^{-2}$] | mean | 179.7 | 176.3 | 129.5 | 163.9 | 146.2 |
| | | SD | 114.5 | 95.9 | 114.4 | 85.0 | 105.2 |
| | | $\rho(LE_{LY},LE_{EC\_c})$ | 0.913 | 0.887 | 0.980 | 0.936 | 0.896 |
| $LE_{EC\_a}$ | [Wm$^{-2}$] | mean | 185.5 | 175.5 | 113.7 | 167.1 | 148.3 |
| | | SD | 110.1 | 89.8 | 115.4 | 84.2 | 104.3 |
| | | $\rho(LE_{LY},LE_{EC\_a})$ | 0.915 | 0.889 | 0.982 | 0.936 | 0.898 |
| $LE_{LY}$ | [Wm$^{-2}$] | mean | 184.3 | 173.4 | 113.7 | 167.3 | 149.9 |
| | | SD | 118.2 | 104.1 | 118.1 | 88.9 | 115.3 |

One may note that $F1$ has the lowest evaporation rate among the humid stations. This will influence the following results throughout.

**Table 2b: Basic evaporation characteristics for the Majadas stations (ρ = correlation coefficient)**

| | | *M1* | *M2* | *M3$_{rainy}$* | *M4* | *M4$_{SM\_moist}$* | *M4$_{SM\_dry}$* |
|---|---|---|---|---|---|---|---|
| $LE_{EC\_o}$ [Wm$^{-2}$] | *mean* | 69.1 | 92.7 | 41.0 | 100.0 | 165.2 | 59.1 |
| | *SD* | 77.0 | 64.1 | 31.1 | 81.8 | 69.8 | 59.2 |
| | *ρ(LE$_{LY}$,LE$_{EC\_o}$)* | 0.928 | 0.867 | 0.771 | 0.910 | 0.723 | 0.943 |
| $\varepsilon$ [Wm$^{-2}$] | *mean* | 125.78 | 133.58 | 122.41 | 161.62 | 181.12 | 149.40 |
| | *SD* | 52.39 | 54.52 | 51.56 | 60.21 | 72.26 | 47.40 |
| $LE_{EC\_c}$ [Wm$^{-2}$] | *mean* | 110.5 | 160.6 | 64.3 | 181.0 | 304.0 | 99.1 |
| | *SD* | 120.0 | 99.4 | 35.5 | 130.3 | 97.1 | 85.5 |
| | *ρ(LE$_{LY}$,LE$_{EC\_c}$)* | 0.957 | 0.926 | 0.803 | 0.967 | 0.898 | 0.959 |
| $LE_{EC\_a}$ [Wm$^{-2}$] | *mean* | 105.4 | 152.6 | 69.6 | 177.1 | 301.9 | 96.8 |
| | *SD* | 104.2 | 92.0 | 35.0 | 132.8 | 91.7 | 87.2 |
| | *ρ(LE$_{LY}$,LE$_{EC\_a}$)* | 0.960 | 0.930 | 0.807 | 0.969 | 0.913 | 0.959 |
| $LE_{LY}$ [Wm$^{-2}$] | *mean* | 103.6 | 153.3 | 68.9 | 177.0 | 300.8 | 99.9 |
| | *SD* | 110.3 | 99.1 | 42.2 | 137.8 | 101.5 | 94.3 |

## 3.2 Differences between means and standard deviations of *LY* and *EC* measurements

Tables 3a and 3b show the absolute differences and their standard deviation between the *EC* data presented in Tables 2a and 2b and *LY* measurements. They indicate how the differences between *LY* and *EC* measurements mostly (except for *F1*) get
 smaller from observed *(D$_o$)* to adjusted values of *LE$_{LY}$ (D$_a$)*.

**Table 3a: Parameter differences (*LY–EC*) for humid stations**

| Parameter | | G1 | G2 | F1 | F2 | RHB |
|---|---|---|---|---|---|---|
| $D_o$ [Wm$^{-2}$] | mean | 31.12 | 24.32 | 6.41 | 33.94 | 10.63 |
| | SD | 18.62 | 25.85 | 23.06 | 15.58 | 14.60 |
| $D_c$ [Wm$^{-2}$] | mean | 5.05 | -3.10 | -15.75 | 3.35 | 3.70 |
| | SD | 3.71 | 8.24 | 3.71 | 3.90 | 10.07 |
| $D_a = d_{rand}$ [Wm$^{-2}$] | mean | -0.98 | -1.34 | 0.67 | 0.18 | 1.60 |
| | SD | 8.06 | 14.33 | 2.70 | 4.73 | 10.94 |


**Table 3b: Parameter differences (*LY-EC*) for Majadas station; semi-arid**

| Parameter | | M1 | M2 | M3$_{rainy}$ | M4 | M4$_{SM\_moist}$ | M4$_{SM\_dry}$ |
|---|---|---|---|---|---|---|---|
| $D_o$ [Wm$^{-2}$] | mean | 34.47 | 60.62 | 27.91 | 77.18 | 135.58 | 40.73 |
| | SD | 33.29 | 34.99 | 11.19 | 55.99 | 31.69 | 35.06 |
| $D_c$ [Wm$^{-2}$] | mean | -6.92 | -7.29 | 4.61 | -0.74 | -3.20 | 0.74 |
| | SD | -9.47 | -0.25 | 6.78 | 7.49 | 4.32 | 8.76 |
| $D_a = d_{rand}$ [Wm$^{-2}$] | mean | -1.81 | 0.70 | -0.75 | 1.47 | -1.16 | 3.08 |
| | SD | 6.02 | 7-08 | 7.22 | 5.06 | 9.73 | 7.13 |


For all stations, the $D_o$-averages are positive, i.e. the *LY* observations are higher on average than the *EC* observations. For the humid stations *F1* and *RHB* the $D_o$ deviations are below the measurement accuracy. The $D_c$ and $D_a$ values are all below the measurement accuracy (except for *F1* in $D_c$) for the humid as well as the semi-arid stations.

## 3.3 Parameters obtained by the *LY-EC* comparison

Tables 4a and 4b present the parameters $d$ (intercept = systematic deviation), $\varepsilon_{red}/\varepsilon$ and $w_{LE}$ as obtained by applying Eq. (3). The systematic deviations means $d$ between *LY* and *EC* are all within the measurement accuracy of *LY* with around $\pm$ 7 Wm$^{-2}$, respectively $\pm$ 20 Wm$^{-2}$ except for *F1* and (marginally) *M2*.

**Table 4a: Parameters for humid stations**

| Parameter | | G1 | G2 | F1 | F2 | RHB |
|---|---|---|---|---|---|---|
| $d$ (intercept) [Wm$^{-2}$] | mean | 6.03 | 1.75 | -16.42 | 3.17 | 2.11 |
| | SD | 7.02 | 9.25 | 6.55 | 3.47 | 5.23 |
| $\varepsilon_{red}/\varepsilon$ | mean | 0.616 | 0.759 | 0.686 | 0.649 | 0.688 |
| | SD | 0.079 | 0.151 | 0.114 | 0.033 | 0.168 |
| $w_{LE}$ | mean | 0.384 | 0.241 | 0.314 | 0.351 | 0.312 |
| | SD | 0.079 | 0.151 | 0.114 | 0.033 | 0.168 |


**Table 4b: Parameters for Majadas station; semi-arid**

| Parameter | | M1 | M2 | M3$_{rainy}$ | M4 | M4$_{SM\_moist}$ | M4$_{SM\_dry}$ |
|---|---|---|---|---|---|---|---|
| $d$ (intercept) [Wm$^{-2}$] | mean | -5.11 | -8.00 | 5.36 | -2.21 | -2.05 | -2.34 |
| | SD | 17.90 | 12.02 | 4.43 | 12.31 | 15.30 | 4.64 |
| $\varepsilon_{red}/\varepsilon$ | mean | 0.678 | 0.506 | 0.809 | 0.515 | 0.230 | 0.726 |
| | SD | 0.282 | 0.222 | 0.039 | 0.290 | 0.079 | 0.182 |
| $w_{LE}$ | mean | 0.322 | 0.494 | 0.191 | 0.485 | 0.770 | 0.274 |
| | SD | 0.282 | 0.222 | 0.039 | 0.290 | 0.079 | 0.182 |


## 3.4 Reduction of the *LY-EC* differences by adjustment expressed in percentages

Tables 5a and 5b give the average and standard deviation differences between *LY* and *EC* values as expressed in percentages of *LY*. The improvements are made visible by comparing the differences before and after adjustments. As such, they may also
be compared to the quotations in Chavez and Howell (2009), Ding et al. (2010) and Evett et al. (2012).


**Table 5a: Comparison of the *LY-EC* differences (means: upper 2 lines; Standard deviations: lower 2 lines) before and after adjustment of the *EC* values, humid**


| adjustment | calculation | G1 | G2 | F1 | F2 | RHB |
|---|---|---|---|---|---|---|
| before | $100*\mathrm{mean}(LE_{LY}-LE_{EC\_o})/\mathrm{mean}(LE_{LY})$ [%] | 16.9 | 14.0 | 5.6 | 20.3 | 7.1 |
| after | $100*\mathrm{mean}(LE_{LY}-LE_{EC\_a})/\mathrm{mean}(LE_{LY})$ [%] | -0.5 | -0.8 | 0.6 | 0.1 | 1.1 |
| before | $100*[\mathrm{SD}(LE_{LY})-\mathrm{SD}(LE_{EC\_o})]/\mathrm{SD}(LE_{LY})$ [%] | 15.8 | 24.8 | 19.5 | 17.5 | 12.7 |
| after | $100*[\mathrm{SD}(LE_{LY})-\mathrm{SD}(LE_{EC\_a})]/\mathrm{SD}(LE_{LY})$ [%] | 6.8 | 13.8 | 2.3 | 5.3 | 9.5 |

**Table 5b: Comparison of the *LY-EC* differences (means: upper 2 lines; Standard deviations: lower 2 lines) before and after adjustment of the *EC* values, Majadas**

| adjustment | calculation | M1 | M2 | M3$_{rainy}$ | M4 | M4$_{SM\_moist}$ | M4$_{SM\_dry}$ |
|---|---|---|---|---|---|---|---|
| before | $100*\mathrm{mean}(LE_{LY}-LE_{EC\_o})/\mathrm{mean}(LE_{LY})$ [%] | 33.3 | 39.5 | 40.5 | 43.6 | 45.1 | 40.8 |
| after | $100*\mathrm{mean}(LE_{LY}-LE_{EC\_a})/\mathrm{mean}(LE_{LY})$ [%] | -1.7 | 0.5 | -1.1 | -0.8 | -0.4 | 3.1 |
| before | $100*\mathrm{SD}(LE_{LY}-LE_{EC\_o})/\mathrm{SD}(LE_{LY})$ [%] | 30.2 | 35.3 | 26.5 | 40.6 | 31.2 | 37.2 |
| after | $100*\mathrm{SD}(LE_{LY}-LE_{EC\_a})/\mathrm{SD}(LE_{LY})$ [%] | 5.5 | 7.1 | 17.1 | 3.7 | 9.6 | 7.6 |


## 3.5 Differences between *LY* and observed, corrected and adjusted *EC* measurements averaged for daytime-hours.

Figures 3a and 3b show the mean daytime cycle of observed hourly differences $LE_{LY} - LE_{EC\_o}$ (denoted as $D_o$ in Tables 3a and

3b) at the individual stations. The averaged $D_o$ differences appear low for the humid data sets and declining towards the afternoon. The Majadas-observations are higher and show a tendency of peaks around noon for the dry season.

Figures 4a and 4b give the corresponding differences between *LY* and corrected *EC* measurements, i.e. $D_c = LE_{LY} - (LE_{EC\_o} + w_{LE}\varepsilon)$.

Figures 5a and 5b demonstrate the $D_a$ values as differences between *LY* and adjusted *EC* measurements ($LE_{EC\_a}$), respectively the random deviations $d_{rand}$. The $D_a$ differences for all stations are mostly within the *LY* measurement accuracy of $\pm$ 7 Wm$^{-2}$, respectively of $\pm$ 20 Wm$^{-2}$ and may be neglected.

## 3.6 Systematic deviations averaged for daytime-hours

Figures 6a and 6b present the systematic deviations $d$ between $LE_{LY}$ and $LE_{EC\_o}$. The systematic deviations for the humid stations are mostly within the $LY$ measurement accuracy of $\pm 7$ Wm$^{-2}$, respectively of $\pm 20$ Wm$^{-2}$ and can thus be neglected for *F2*, *G2*, *RHB*, *M3* and *M4*. For *F1* the deviations are exceeding the measurement accuracy quite substantially throughout the daytime period, while the deviations at *G1* are larger only in the morning and afternoon and at *M1* and *M2* from noon until the evening.

## 3.7 Averaged hourly daytime values for $w_{LE}$

Figures 7a and 7b show the mean course of $w_{LE}$ during daytime-hours using the average of all $w_{LE}$ values at a specific hour. The number of bins used in Fig. 7a per station varies from 6 (*F1*), 8 (*F2, G2, RHB*) to 14 (*G1*). The number of bins used for Majadas in Fig. 7b varies from 5 to 12, depending on the used period. We distinguish between the drying periods (about March to August) in red and yellow as well as the one "rainy" period *M3* (end of August 2017 to beginning of January 2018) in blue. Figure 7b also splits *M4* into a period with "high soil moisture" (20.04. to 23.06., yellow line with blue triangles) and a "low soil moisture" (01.07. to 04.09., yellow line with yellow triangles). Both periods are under high temperatures and very sparse rainfall. For soil moisture, see Fig. 8b.

All humid averaged values of daytime-hours of $w_{LE}$ are roughly within the range of around 0.2 and 0.4. Their standard deviation is highest in the hours around noon (not shown), which relates to the fact that the absolute differences between $LE_{LY}$ and $LE_{EC}$ observations are comparably small during stable to weakly unstable conditions in the morning and evening. For Majadas, variations in the various datasets are higher, especially for the drying period (i.e. no rainfall, but still high soil moisture) of *M4* (topmost line in Fig. 7b).

## 3.8 Temporal patterns

### 3.8.1 $w_{LE}$ in time

Figures 8a and 8b show two different situations for the development of $w_{LE}$ in time under varying soil moisture. While Fig. 8a presents a limited dry period under humid conditions (*G1*), Fig. 8b demonstrates a gradually drying situation over 212 days (20.04. to 04.09. 2018) for *M4*.

### 3.8.2 *LY-EC* deviations in time

Figures 9a and 9b illustrate the *EC* deviations from the *LY* values before (light green) and after (blue) *EC* adjustments along the analyzed time period for *F2* (7a) and *M4* (7b). They demonstrate again the remaining high variation.

**4. Discussion**

The method applied offers two results: (1) corrected $LE_{EC\_c}$ values as given by $LE_{EC\_c} = LE_{EC\_o} + w_{LE}\,\varepsilon$ and (2) adjusted $LE_{EC\_a}$ values as given by $LE_{EC\_a} = LE_{EC\_c} + d$. One may consider $LE_{EC\_c}$ as *weakly linked* to the *LY* measurements via the *wL*-regression and $LE_{EC\_a}$ as *strongly linked* to $LE_{LY}$ via both $w_{LE}$ as well as $d$. Differences between the two mostly range within
the measurement accuracies (Tables 3a and 3b).

In general, *LY* measured data are higher than data based on the *EC* method. This is in accordance to literature (e.g. Chavez and Howell, 2009). They differ substantially less in humid climate with around 10 to 30 Wm$^{-2}$ (0.35 to 1.0 mm d$^{-1}$) than at Majadas station with around 30 to 60 Wm$^{-2}$ (1.0 to 2.1 mm d$^{-1}$).


The adjustment of the *EC* to the *LY* data expressed by the differences $D_a$ hint at a nearly perfect match for the means (Tables 3a and 3b). They are all in the range of the measurements accuracies. All standard deviations given by the difference SD($LE_{LY}$) – SD($LE_{EC\_a}$), respectively SD($LE_{EC\_c}$), increase with adjustments, but remain less than SD($LE_{LY}$) (see SD for $D_o$ and $D_a$ values in Tables 3a and 3b). The difference between SD($LE_{LY}$) and SD($LE_{EC\_o}$) is getting bigger, since SD($LE_{EC\_o}$) gets smaller after
correction, whereas SD(LE$_{LY}$) remains the same.

The effectiveness of our method is demonstrated by comparing our results given in Tables 5a and 5b with the following results achieved by former authors:
- Chavez and Howell (2009) with reductions of *LY-EC* differences from -28.8 % to 6.2 %, respectively from -26.0 % to -
12.3 %, with an accuracy of $\approx 0.9 \pm 14$ Wm$^{-2}$, respectively $\approx - 2.8 \pm 11$ Wm$^{-2}$
- Evett et al. (2012), mentioning $LE_{EC}$ measurements errors within $\approx 55$ to 78 Wm$^{-2}$, which were reduced after forced closure of the energy gap to $LE_{LY}$ and $LE_{EC}$ differences between 17.4 and 18.7 % and
- Ding et al. (2010), quoting that differences between *LY* and *EC* measurements could be reduced from -22.4 % to -6.2 %.

It surprises that the systematic deviations *d* between *LY* and *EC* measurements (Tables 4a and 4b) are on average within the measurement accuracy with exception of *F1* and (marginally) *M2*. For the humid regions *d* is positive (4 cases) as well as negative (1 case). For Majadas *d* is positive only for *M3,* measured during rainy season. For *M4* the *d* values are distinctly

below measurement accuracy (Table 4b; Fig. 6b). One could expect a more pronounced difference of $d$ for the two different measurements devices ($RHB$ and lower boundary-controlled lysimeters).

The energy gaps are in the range of 25 to 100 Wm$^{-2}$ for the humid stations. They are much higher for Majadas with around 120 to 180 Wm$^{-2}$. The gaps reduce to about 50 to 80 % after partial energy closure. They appear rather constant (around 70%) for the humid regions and vary more for Majadas, for which the most striking variations, i.e. 23% and 72.6% respectively, occur with $M4$ during high and low soil moisture (Tables 4a and 4b, lines $\varepsilon_{red}$).

The calculated $w_{LE}$ values appear nearly independent of daytime-hours (Fig. 7a and 7b). Data from humid climate gave hourly averaged $w_{LE}$ values within a surprisingly narrow range of 0.2 to 0.4. The corresponding values for Majadas show wider variations. During the non-rainy-season, they differ more substantially for $M4$ with high soil moisture ($w_{LE}$ around 0.78) and low soil soil moisture ($w_{LE}$ around 0.25). This discrepancy of $w_{LE}$ is mitigated by extending the daily time window of the Majadas data (Section 2.4).

Standard deviations of $w_{LE}$ for daytime-hours averages change little, with a tendency of smaller values in the morning and evening. This relates to small absolute values of evaporation during stable or weakly unstable conditions.

The value of $w_{LE}$ seems partly positively correlated to the magnitude of evaporation. This correlation is indicated in Fig. 8b, where a drop in $w_{LE}$ follows $LE_{EC\_c}$.

We could not find any explanation for the unexpected drop of $d$ values for $G2$ (Fig. 6a).

## 5. Summary and conclusions

The applied partial closure gives, according to our knowledge, for the first time a fully rational method to partially close the energy gap and a more detailed description of the correlations between $LY$ and $EC$ observations. The method gives two results for improved $LE_{EC}$ estimates, one weakly linked and one strongly linked to the $LE_{LY}$ readings. Their differences appear negligible in view of the inaccuracies of the input data. The method also allows a distinction between systematic and random deviations for the first time, probably. The $w_{LE}$ weight-averages are rather stable during daytime. The systematic deviations and random deviations (Tables 4a and 4b) are mostly below or very close to measurements accuracies.

For the future, one should try to increase the information of $LY$ as well as $EC$ measurements. In a first step we recommend to perform the comparison of $LY$ and $EC$ measurements based on 5 to 10 minutes lysimeter intervals, and center the one/half-hourly averaging window accordingly on the $EC$ raw data. We expect an improvement of the accuracy of $w_{LE}$, $d$ and $d_{rand}$ estimates thereby. The benefit of using more highly resolved lysimeter data is described in Ruth et al. (2018).

In long terms, one may think of improving measurement accuracies of relevant input data. Lysimeter measurements should include negative values (condensation) and consider the influence of wind. The former can be realized by including rain observation on a high temporal scale to identify a mass increase in the absence of rain, i.e., dew formation (Ruth et al.; 2018). If a high-precision lysimeter capable of resolving evaporation as well as condensation is available complementary to an *EC* set-up, *LE* can directly be obtained from the lysimeter. As long as no improvements are realized, as a pragmatic solution for

full energy balance closure we recommend closing by attributing one third of the gap $\varepsilon$ to each of the three weights. This is common practice in land surveying. This recommendation is supported by the fact that we found generally rather constant $w_{LE}$ values during daytime between 0.2 and 0.4.

We recommend to test also high-quality flag 0 datasets (Mauder et al, 2013) for plausibility by the Out-of-Bound method, which may be derived from Wohlfahrt and Widmoser, 2013.

The method proposed here may also be applied if reliable sap flow measurements are available instead of lysimeter observations. We guess that an adoption of our method may apply to partial energy closure by heat fluxes if surface temperatures estimates are known from telemetry.

## 6. Data availability

The data basis for the presented analyses is available at https://doi.org/10.5281/zenodo.3957208 (Graswang and Fendt 2013-2014), https://doi.org/10.3929/ethz-b-000420733 (Rietholzbach 2013) and https://doi.org/10.5281/zenodo.3964082 (Majadas 2016-2018). The datasets consist of the half-hourly or hourly, respectively, time series of lysimeter and eddy covariance evaporation, as well as ancillary data described in the text.

## Author contributions

Peter Widmoser initiated the study, conducted the analyses and wrote a first version of the manuscript. Dominik Michel revised the article and put it into shape for publication.

## Competing interests

The authors declare that they have no conflict of interest.

## Acknowledgements

We are grateful to M. Mauder and R. Kiese, Institute of Technology, KIT, Germany, O. Perez-Priego, Max Planck Institute for Biogeochemistry, Germany and S. I. Seneviratne and M. Hirschi, Institute for Atmospheric and Climate Science, ETH

Zurich, Switzerland for the data as well as G. Wohlfahrt, Institute for Ecology, University of Innsbruck, Austria, for every support. The sites Graswang and Fendt are part of the TERENO observatory, which is funded by the Helmholtz Association and the Federal Ministry of Education and Research. Majadas lysimeters data were supported by the Alexander von Humboldt Foundation that supported the research with the Max-Planck Prize to Markus Reichstein. For the data collection we thank Arnaud Carrara (CEAM, Valencia), Oscar Perez-Priego, Tarek El-Madany, Olaf Kolle, and Mirco Migliavacca (Max Planck Institute for Biogeochemistry).

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

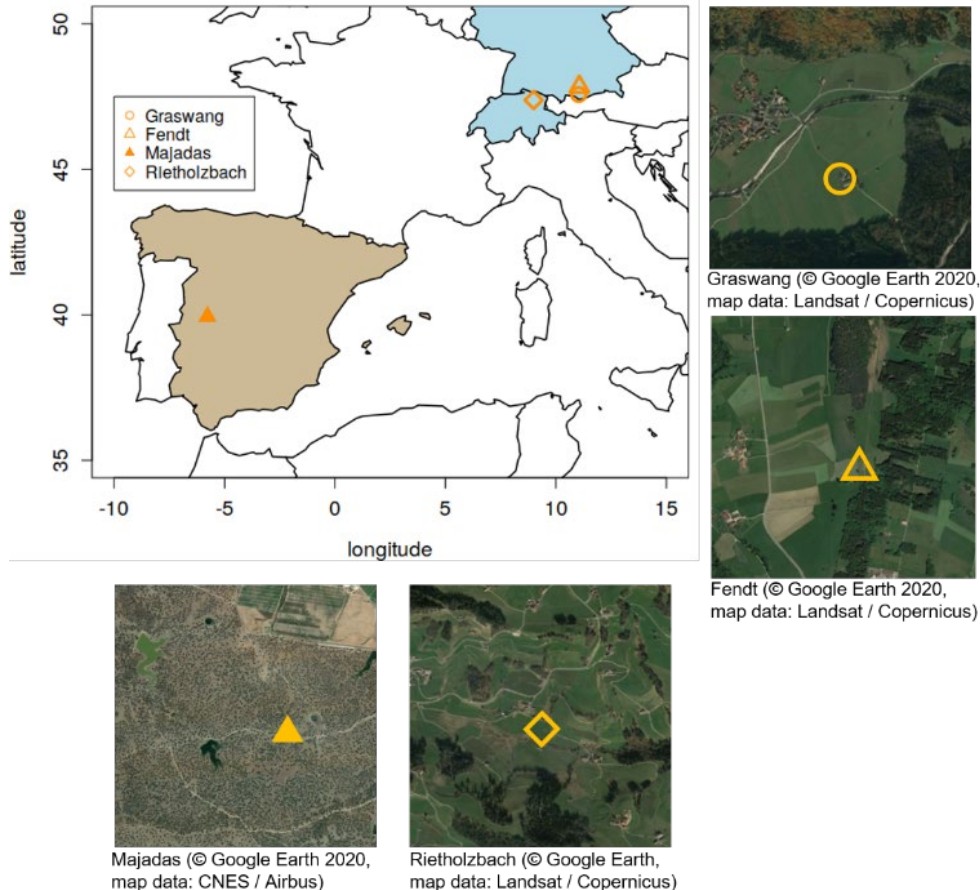

Graswang (© Google Earth 2020, map data: Landsat / Copernicus)

Fendt (© Google Earth 2020, map data: Landsat / Copernicus)

Majadas (© Google Earth 2020, map data: CNES / Airbus)

Rietholzbach (© Google Earth, map data: Landsat / Copernicus)

**Fig. 1: Location and satellite view of the used stations and their surrounding area. The symbols denote the location of the lysimeters.**

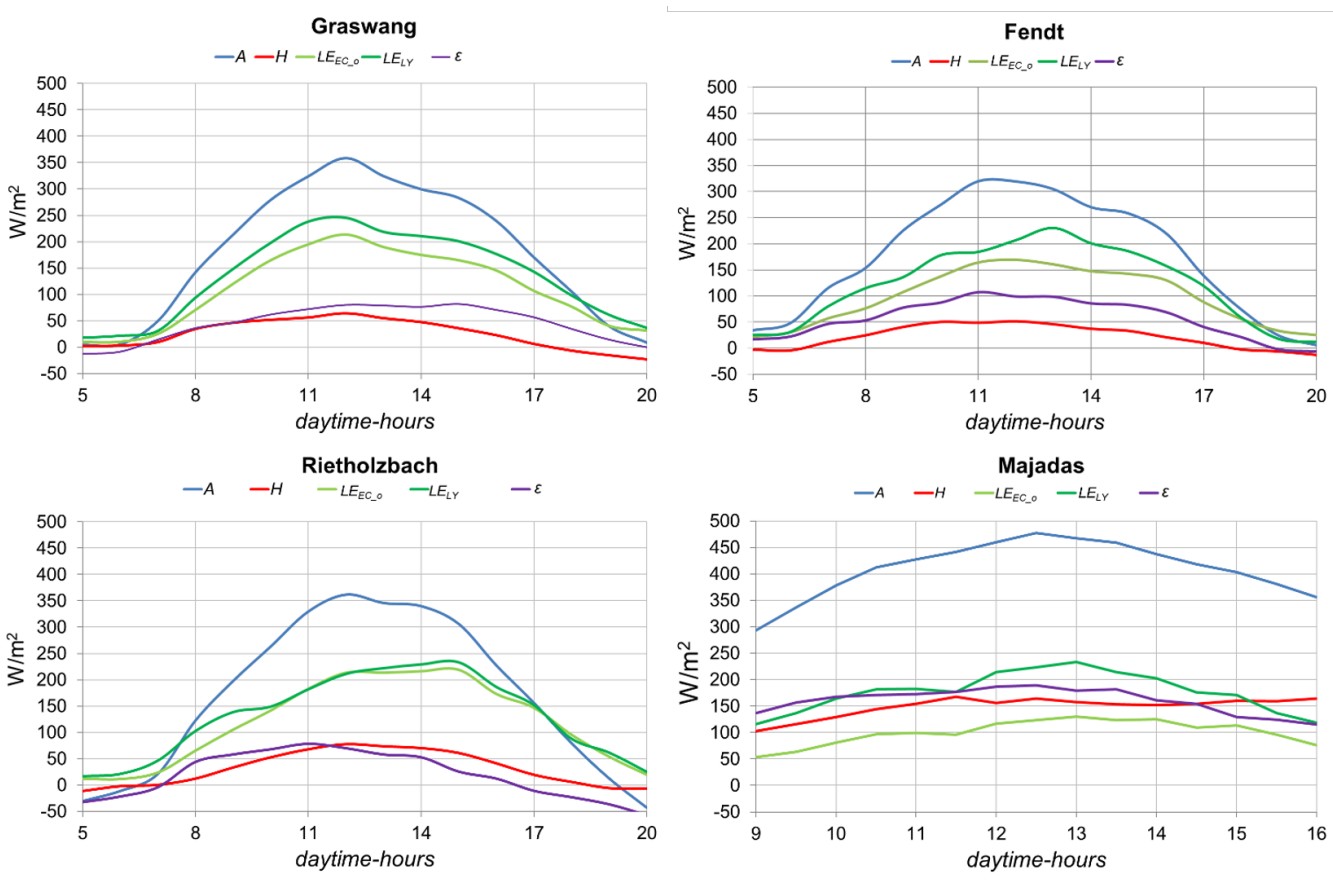

**Fig. 2: Average daytime course of available energy (*A*), sensible heat flux (*H*), *EC*-based (*LE_{EC_o}*) and lysimeter-based (*LE_{LY}*) latent heat flux and the energy gap (*ε*) at the four stations. Note that for Majadas the diurnal cycle represents the dry season (*M4*).**


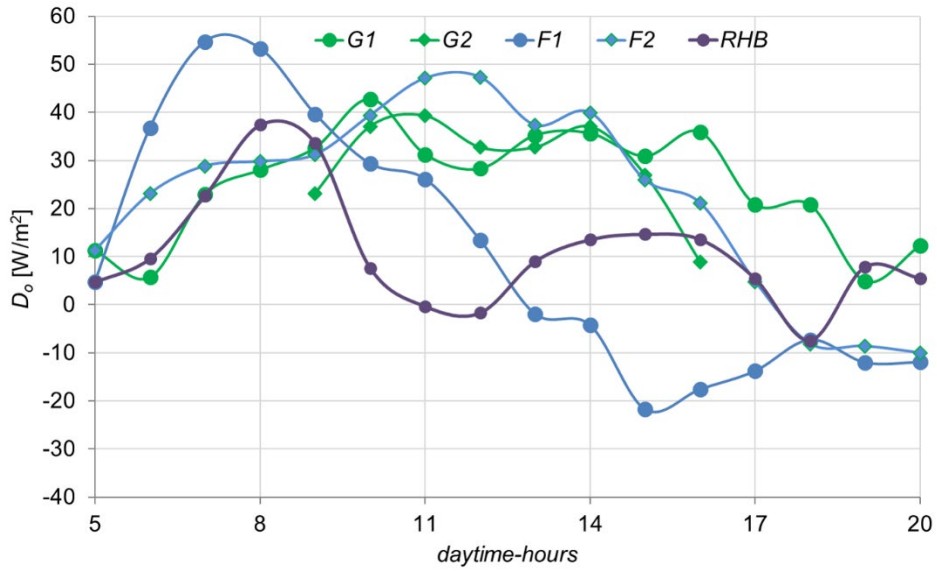

555        **Figure 3a:** $D_o = LE_{LY} - LE_{EC\_o}$ **as a function of daytime-hours; humid.**

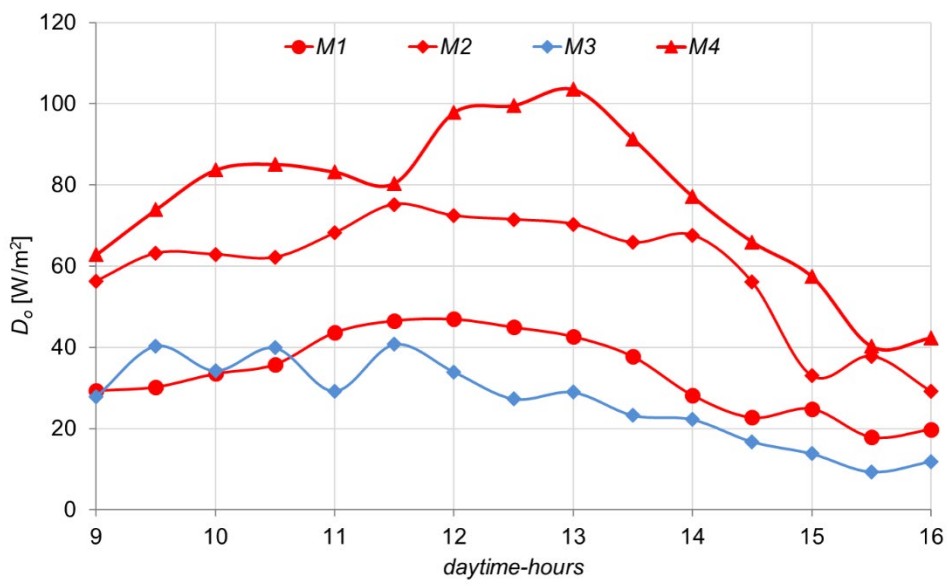

**Figure 3b:** $D_o = LE_{LY} - LE_{EC\_o}$ **as a function of daytime-hours; Majadas; red: dry; blue: wet season.**


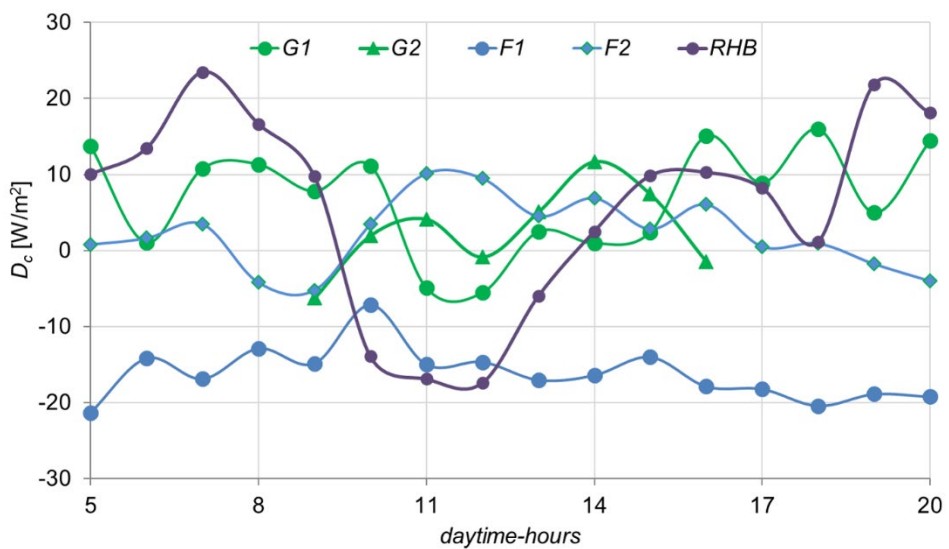

565             **Figure 4a: Differences $D_c = LE_{LY} - LE_{EC\_c}$; humid.**

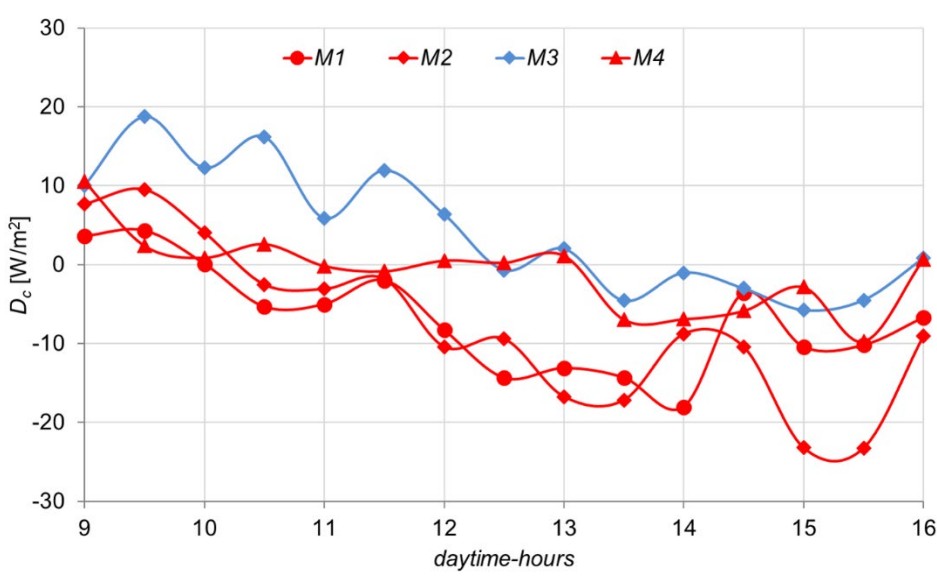

**Figure 4b: Differences $D_c = LE_{LY} - LE_{EC\_c}$; Majadas red: dry; blue: wet season.**


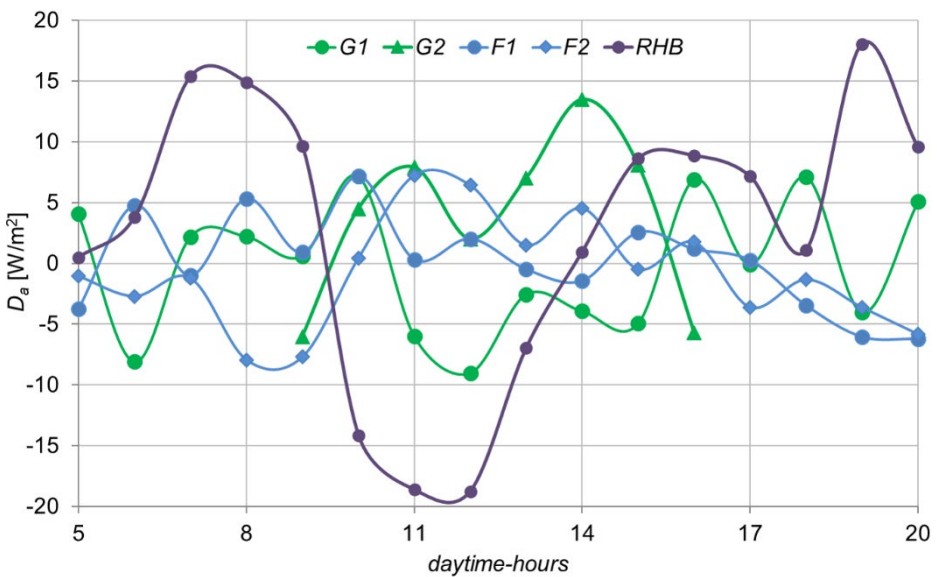

**Figure 5a: Differences $D_a$ between $LE_{LY}$ and $LE_{EC\_a}$ values as a function of daytime-hours; humid.**


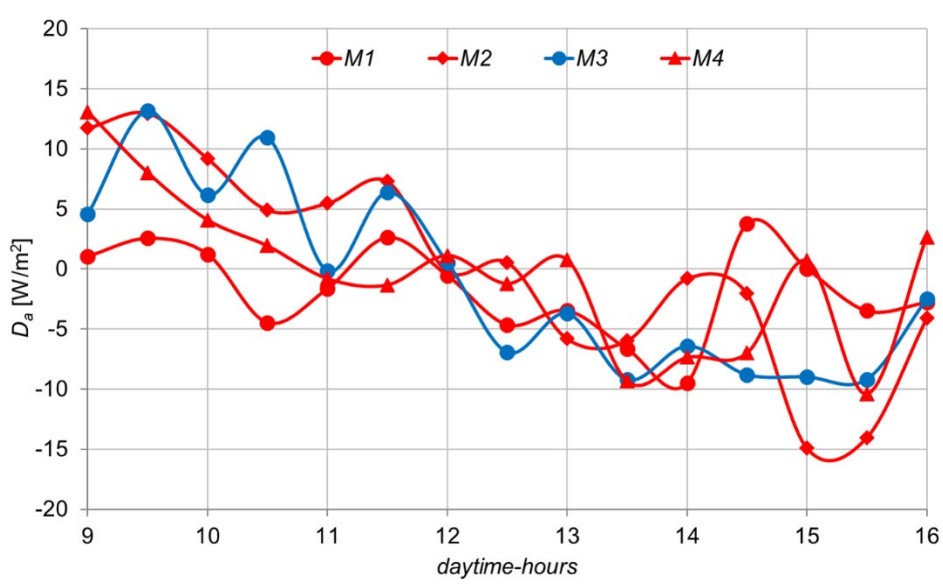

**Figure 5b: Differences $D_a$ between $LE_{LY}$ and $LE_{EC\_a}$ values as a function of daytime-hours, Majadas; red: dry; blue: rainy season.**


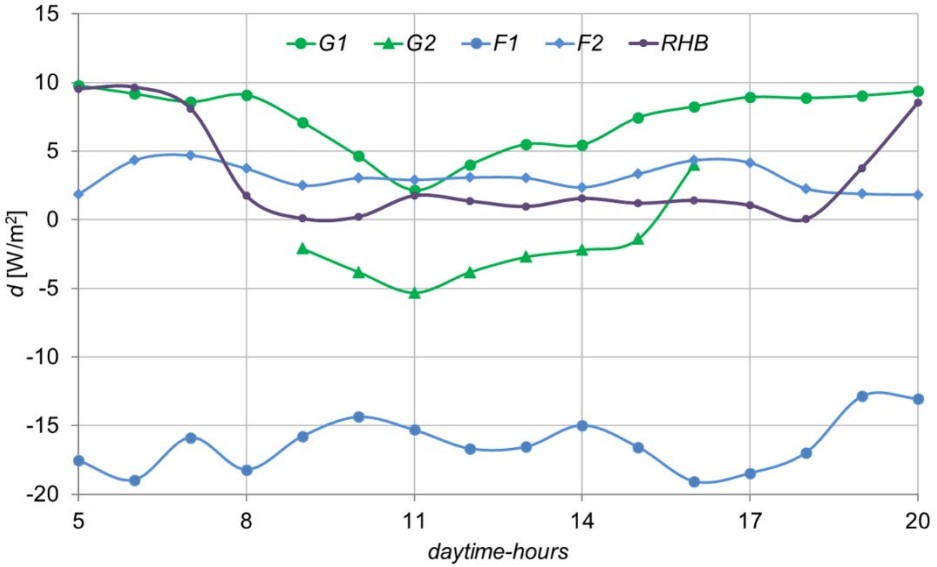

**Figure 6a: Systematic differences $d$ between $LE_{LY}$ and adjusted $LE_{EC\_a}$; humid.**


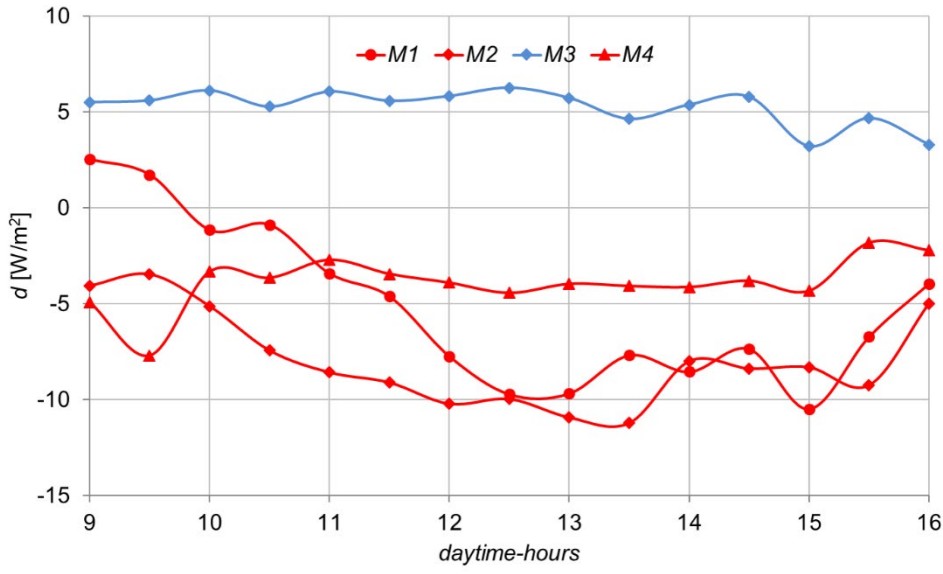

**Figure 6b: Systematic differences $d$ between $LE_{LY}$ and adjusted $LE_{EC\_a}$; Majadas red: dry season; blue: rainy season.**

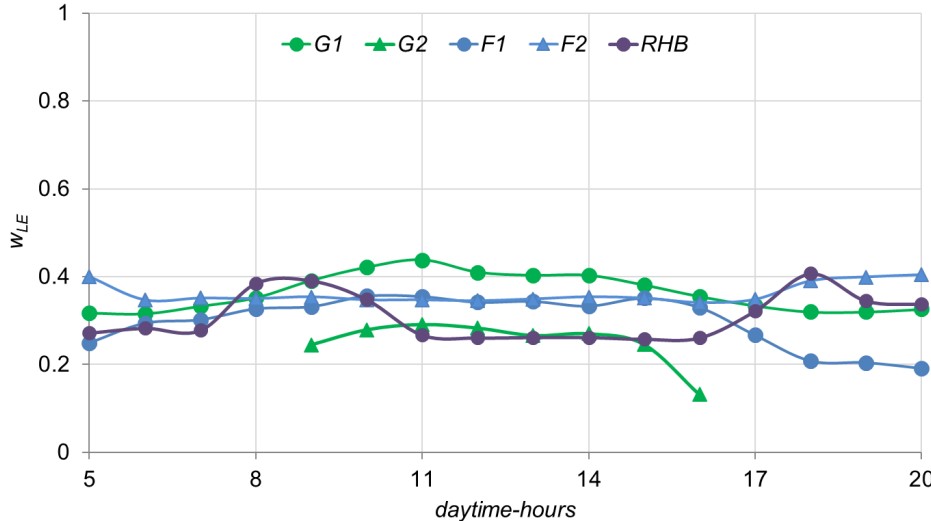


**Figure 7a: Averaged daytime-hours values for *LE*-weights *w$_{LE}$*; humid.**

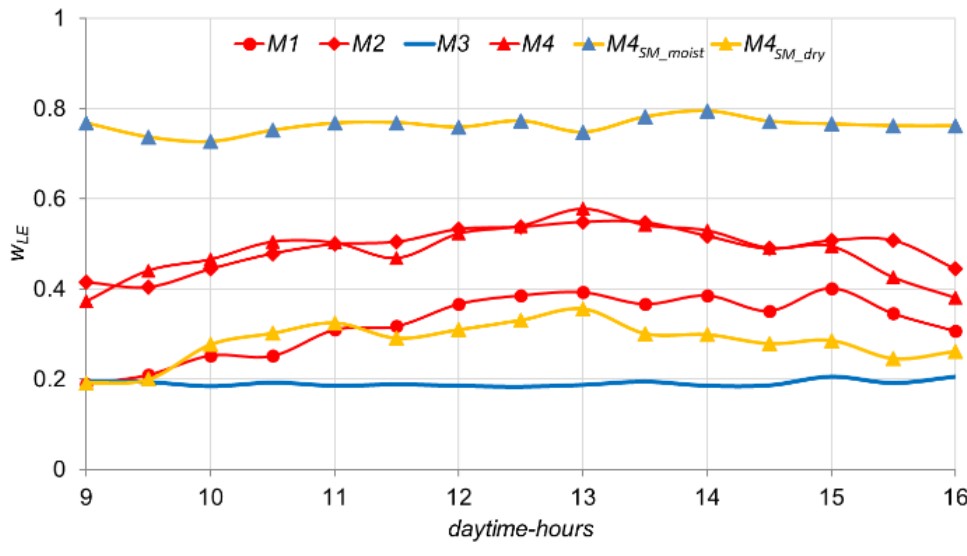

**Figure 7b: Averaged values daytime-hours for *LE*-weights (*w$_{LE}$*) in Majadas red: dry; blue: wet season. *M4* split into the period**
**"high soil moisture" (20.04. to 23.06., yellow line, blue triangles) and "low soil moisture" (01.07. and 04.09., yellow line, yellow triangles).**


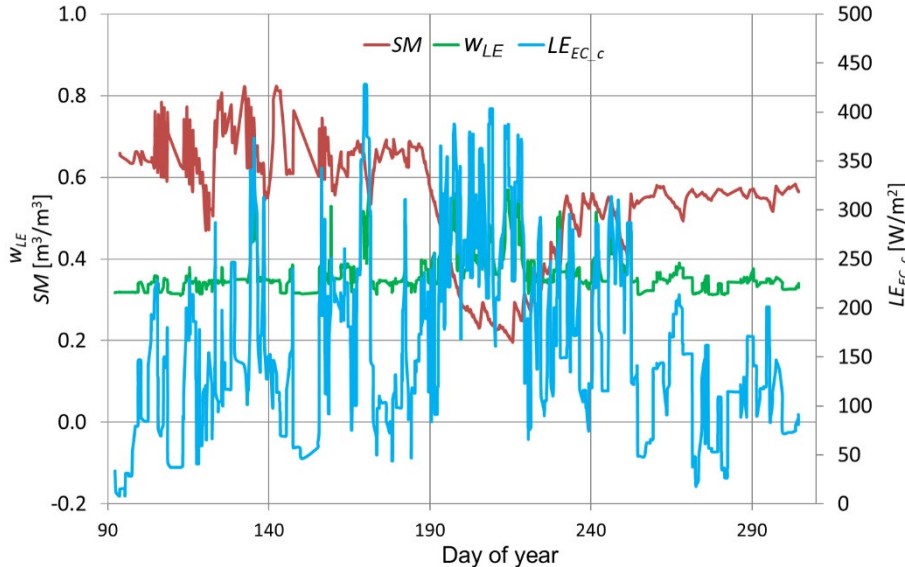

**Figure 8a: Development of $w_{LE}$ (smoothed, dark green), $LE_{EC\_c}$ (smoothed, blue) and soil moisture (brown) including a dry spell in 2013 for *G1*, humid. All data shown are measured from 9 am to 4 pm. A moving median filter with a window length of 11 hours was used for smoothing the $w_{LE}$ and *LE* data.**


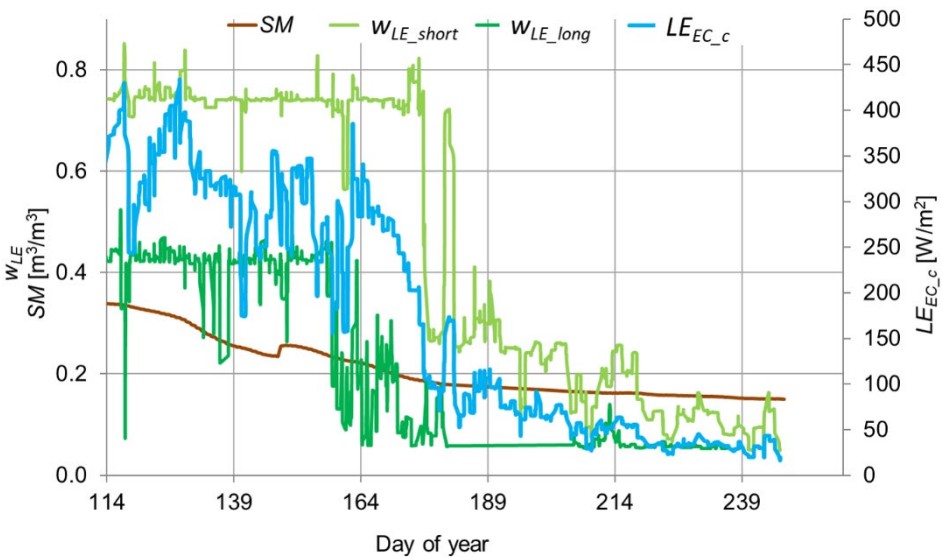

Figure 8b: Development of $w_{LE}$ (smoothed, light green) results from values measured between 9 am and 4 pm, the lower $w_{LE}$ (smoothed, dark green) results from estimates from 5 - 9 am and 4 - 8 pm and measurements between 9 am - 4 pm (see Section 2.4), and corrected $LE_{EC\_c}$ (smoothed, blue) along with soil moisture (SM, brown) from 21.04. to 04.09. 2018 for $M4$, semi-arid. A moving median filter with a window length of 11 hours was used for smoothing the $w_{LE}$ and $LE$ data.

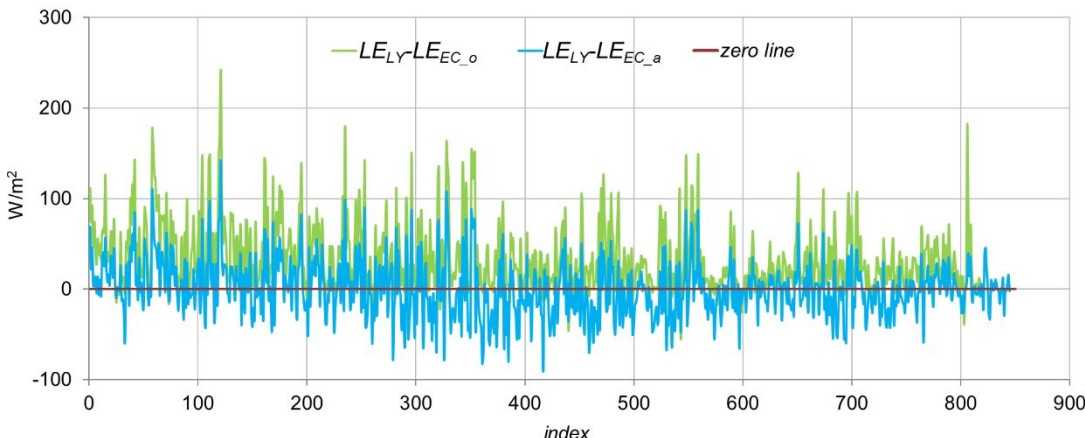

Figure 9a: $EC$ deviations from $LY$ observations before (green) and after (blue) $EC$ adjustments along observation period for station $F2$.


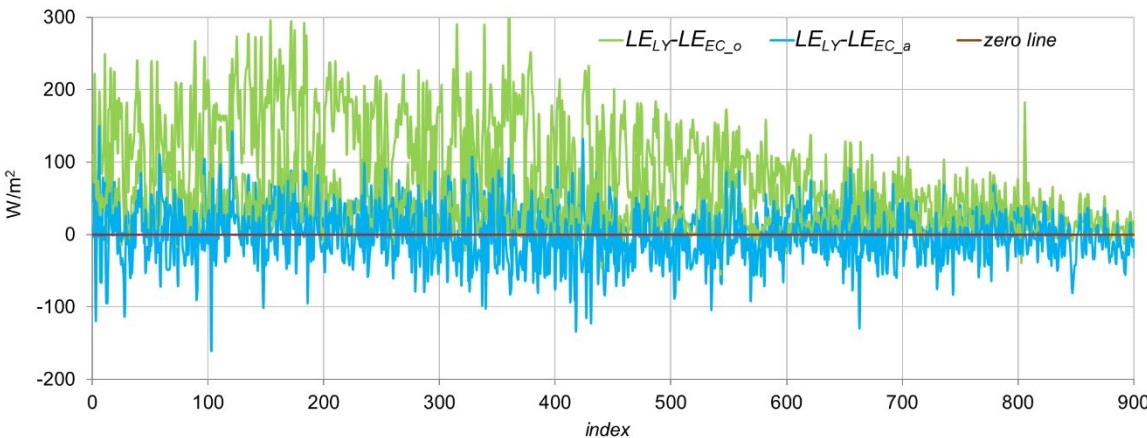

**Figure 9b:** *EC* deviations from *LY* observations before (green) and after (blue) *EC* adjustments along observation period for station *M4*.
