# Peer review of "Partial energy balance closure of eddy covariance evaporation measurements using concurrent lysimeter observations over grassland"

_Hydrology and Earth System Sciences, 2020_

## Referee Comment (RC1) · Anonymous Referee #1 · 15 Jul 2020

General comments.

The manuscript is relevant for hydrological studies. The energy balance gap in eddy correlation (EC) measurement is an ongoing topic that deserves attention. The manuscript is well written. It is however rather brief and seems to be primarily readable for insiders. I am personally in favor of these short and concise manuscripts. It has the characteristics of a technical note. The editor could consider to publish it as such.

Major comment.

[Figure]

Throughout the manuscript d is considered to describe a systematic difference be-
tween LY and LE. Lysimeter measurements are systematically larger than EC mea-
surements. However d can be negative or positive for different sites. Stating that d
describes a systematic difference is confusing and not correct. d is simply the intercept
of the linear regression model having the energy balance gap (epsilon) at the xaxis
and LY – LE on the yaxis. If the energy balance gap (epsilon) is zero d will remain.
We could even argue that we could drop d in the model since cLE does already quite
a good job in correcting LE (table 3a in the manuscript). The description of d and the
conclusions based on d should be corrected throughout the manuscript.

Specific comments.

1. The formulation of line 30 to 33 is unclear to me. Was just the difference between LE
measured with EC and LE measured with lysimeters smaller than the energy balance
gap? I would suggest to reformulate this part and explain "reduced the differences".

2. In line 37 partial evaporation closure is mentioned. Shouldn't this be partial energy
balance closure? It is not clear what is meant.

3. In the equations the dimensionless weights are in the form wA, wH, wL. I find
this confusing. I would suggest to use subscripts for A, H and L. Otherwise I could
interpreted wH as a weight times sensible heat, which is not the case.

4. LY is used for lysimeter LE in the equations. This is confusing. I think the notations
should be reconsidered.

5. In line 312 the difference between LE and LY for humid climates is described as
surprisingly little. I think this is not correct. The difference is large. 10 to 30 W/m2 is
similar to 0.35 to 1 mm/d which is equal to 128 to 365 mm/year. On a water balance in
most regions of the world including Europe these differences are large.

6. The formulation of line 320 to 325 is unclear. "The adjustments reached in this paper
are higher". Did the corrections/adjustments lead to better results? How come? If I

am correct the literature citations in these lines have used full energy balance closure techniques with still large differences with lysimeter measurements right? (I haven't checked) This is something different.

7. Line 352. To my opinion better to reformulate this line. The presented manuscript is basically fitting a certain model, but that doesn't tell anything about what is best.

8. In line 359 the authors suggest to use 5 to 10 min resolution lysimeter data. I think this is unrealistic. There are no lysimeters other than dead weight compensated lysimeters that can measure accurately at such a fine resolution. Even presenting data on hourly intervals is to my opinion debatable. I would rather suggest to do the analysis on daily data or the sum of daytime data. The analysis would than be much less affected by lysimeter measurement errors and as proved in manuscript the correction weights for most situations are constant during the day.

Technical corrections.

9. Typo: At the end of line 247 the word "und" should be "and".

10. Figure 7a. Legend item "zero line" should be brown.

---

## Author Comment (AC1) · 28 Jul 2020

We would like to update and specify the acknowledged work that made the current study possible, namely from the Institute of Technology, KIT, Germany and the Max Planck Institute for Biogeochemistry, Germany. Please also note the new links to the individual datasets:

We are grateful to M. Mauder und R. Kiese, Institute of Technology, KIT, Germany, O. Perez-Priego, Max Planck Institute for Biogeochemistry, Germany and S. I. Seneviratne

and M. Hirschi, Institute for Atmospheric and Climate Science, ETH Zurich, Switzerland for the data as well as G. Wohlfahrt, Institute for Ecology, University of Innsbruck, Austria, for every support. The sites Graswang and Fendt are part of the TERENO observatory, which is funded by the Helmholtz Association and the Federal Ministry of Education and Research. Majadas lysimeters data were supported by the Alexander von Humboldt Foundation that supported the research with the Max-Planck Prize to Markus Reichstein. For the data collection we thank Arnaud Carrara (CEAM, Valencia), Oscar Perez-Priego, Tarek El-Madany, Olaf Kolle, and Mirco Migliavacca (Max Planck Institute for Biogeochemistry).

The data basis for the presented analyses is available at https://doi.org/10.5281/zenodo.3957208 (Fendt and Graswang 2013-2014), https://doi.org/10.5281/zenodo.3964082 (Majadas 2016-2018) and https://doi.org/10.3929/ethz-b-000420733 (Rietholzbach 2013). The datasets consist of the half-hourly or hourly, respectively, time series of lysimeter and eddy covariance evaporation, as well as ancillary data described in the text.

---

## Author Comment (AC2) · 5 Aug 2020

We thank referee #1 for the comments. We respond to them below the referee's comment (italic).

General Comment:
*The manuscript is relevant for hydrological studies. The energy balance gap in eddy correlation (EC) measurement is an ongoing topic that deserves attention. The*

[Figure]

*manuscript is well written. It is however rather brief and seems to be primarily readable for insiders. I am personally in favor of these short and concise manuscripts. It has the characteristics of a technical note. The editor could consider to publish it as such.*

The positive feedback on the importance of the topic as well as on the writing is much appreciated. We leave it to the Editor to publish the paper as a technical note.

Major Comment:
*Throughout the manuscript d is considered to describe a systematic difference be-tween LY and LE. Lysimeter measurements are systematically larger than EC measurements. However d can be negative or positive for different sites. Stating that d describes a systematic difference is confusing and not correct. d is simply the intercept of the linear regression model having the energy balance gap (epsilon) at the xaxis and LY – LE on the yaxis. If the energy balance gap (epsilon) is zero d will remain. We could even argue that we could drop d in the model since cLE does already quite a good job in correcting LE (table 3a in the manuscript). The description of d and the conclusions based on d should be corrected throughout the manuscript.*

We changed in line 145 from "the y-intercept (d) can be interpreted" to "might be interpreted".
We will, however, not correct the description of d and conclusions drawn from it. We have the following arguments:
- Our interpretation is correct from a statistical point of view
- Even if the interpretation of d may be irritating in the case of negative d as e.g. for Fendt 2013, we feel this irritation is helpful for a more detailed analysis of the data
- The new method proposed has been applied so far only to four stations. Further experience with data evaluations from more sites may decide about the d-interpretation

Specific Comments:

*1) The formulation of line 30 to 33 is unclear to me. Was just the difference between LE measured with EC and LE measured with lysimeters smaller than the energy balance gap? I would suggest to reformulate this part and explain "reduced the differences".*

This refers to the difference between EC and lysimeter data. The formulation was indeed unclear, as the numbers referred to the case without forced closure, which is confusing with the previous sentence. We reformulated lines 30 to 32 accordingly: After forced closure of the energy balance, the difference between daytime LY and EC data on two fields could be reduced from -28.8% to 6.2%, respectively from -26% to -12.3%, with an accuracy. . .

*2) In line 37 partial evaporation closure is mentioned. Shouldn't this be partial energy balance closure? It is not clear what is meant.*

We corrected in line 56 (not line 37) to "partial energy balance closure".

*3) In the equations the dimensionless weights are in the form wA, wH, wL. I find this confusing. I would suggest to use subscripts for A, H and L. Otherwise I could interpreted wH as a weight times sensible heat, which is not the case.*

We feel it might be useful to apply the weight-notation in agreement with two previous publications in AGRFORMET. We leave it to the Editor to decide whether to change the weight-notations.

*4) LY is used for lysimeter LE in the equations. This is confusing. I think the notations*

*should be reconsidered.*

We change the notation for lysimeter- and EC-values to $LE_{LY}$ and $LE_{EC}$

*5) In line 312 the difference between LE and LY for humid climates is described as surprisingly little. I think this is not correct. The difference is large. 10 to 30 W/m2 is similar to 0.35 to 1 mm/d which is equal to 128 to 365 mm/year. On a water balance inmost regions of the world including Europe these differences are large.*

We changed the sentence on line 312 to: "They differ much less for humid climate with around 10 to 30 Wm$^{-2}$ (0.35 to 1.0 mm d$^{-1}$) than at Majadas-Station with around 30 to 60 Wm$^{-2}$ (1.0 to 2.1 mm d$^{-1}$)".

*6) The formulation of line 320 to 325 is unclear. "The adjustments reached in this paper are higher". Did the corrections/adjustments lead to better results? How come? If I am correct the literature citations in these lines have used full energy balance closure techniques with still large differences with lysimeter measurements right? (I haven't checked) This is something different.*

We replace line 320 by: "The effectiveness of our method is demonstrated by comparing our results given in Table 5 with the following results achieved by former authors:"

*7) Line 352. To my opinion better to reformulate this line. The presented manuscript is basically fitting a certain model, but that doesn't tell anything about what is best.*

We changed the sentence on line 352 to: "…gives, according to our knowledge, for
the first time a fully rational method to partially close the energy gap and a more detailed description of the correlations between LY and EC-observations."

*8) In line 359 the authors suggest to use 5 to 10 min resolution lysimeter data. I think this is unrealistic. There are no lysimeters other than dead weight compensated lysimeters that can measure accurately at such a fine resolution. Even presenting data on hourly intervals is to my opinion debatable. I would rather suggest to do the analysis on daily data or the sum of daytime data. The analysis would than be much less affected by lysimeter measurement errors and as proved in manuscript the correction weights for most situations are constant during the day.*

We can understand the reasoning of the referee. But we still feel it is justifiable to at least propose reduced time intervals. Higher time resolution of LY-data could can easily be retrieved. However, thorough filtering of the data is required for high resolution data, e.g. using the AWAT filter (Peters 2017, https://doi.org/10.1016/j.jhydrol.2017.04.015). As far as our method is concerned, Eq. 3 relies on statistics and its reliability is as such dependent on a high input number.

Technical corrections

*9) Typo: At the end of line 247 the word "und" should be "and".*

We corrected the error in line 247.

*10) Figure 7a. Legend item "zero line" should be brown.*

We changed the color of the zero line in Fig. 7a.

---

## Referee Comment (RC2) · Anonymous Referee #2 · 18 Sep 2020

This topic is very relevant, as surface energy non-closure is still one of the outstanding problems in micrometeorology. Lysimetry may indeed be one of the techniques that can help to shed more light on this fundamental problem. Interestingly, and as a byproduct this technique brings together the worlds of hydrology and micro-meteorology, which is very welcome.

Here follows my main critics, which then will be more detailed in specific points below. At the end more textual comments and suggestions are given.

1) A lot of results are presented in this paper, but often without much comments by the authors. In that respect the paper looks more like a technical report which may form the basis of peer-reviewed paper. I urge the authors to take the reader along the circa eight tables and describe in text what the main message of each table is.

2) Although the authors focus on the relation between LY and EC measurements, they also use the other observations of the surface energy balance, net radiation and surface soil heat flux in essential parts of their analysis. These observations should be described aswell in section 2.

3) A rational for the used method is lacking, given that the authors state that they are mainly interested in the relation of LY and EC evaporation observations.

4) Section 4 is more a summary of results then a discussion. For example the part on standard deviations needs a discussion on what these comparison of SDV means and what can be learned of it. Now there are so many nice statistical results and apparently so little conclusions can be drawn. The question is whether these statistical techniques alone are sufficient to grasp the mechanisms behind the differences observed. Perhaps these should be accompanied by detailed case to case studies.

5) The text is not always as precise at it could be, some examples are given below. But there are more of these occasions. Please copy edit the text carefully on this aspect.

Specific comments

6) L45: How is the evaporation fraction used to correct? Is that different from Bowen ratio preservation?

7) Table 1: It would be nice to have the other information (measurement time interval, vegetation type, period of the day used) also in the table. This may require to turn the table by 90 degrees.

8) What would be the influence of the oak trees at station Majadas on the flux observations.

9) S2.2: Here general error characteristics are given. Are the authors sure that these can applied to the various sites used here. Are there any specific circumstances which may have an influence on the error characteristics. For example, how well are the conditions in the lysimeter kept comparable to the surrounding fields. Are there infrared surface temperature observations to judge possible inhomogeneities between lysimeter and surroundings?

9) S2.3: Here I have the same questions. I find these error estimates to general. It is always good to look at specifics of datasets/sites. EC measurements require all kinds of corrections. I miss a statement on the applied methods, and any differences in treatment per site.

10) L113: Wohlfart and Widmoser (2013) apply the out-of-bound concept for corrected EC observations to judge whether this corrections lead to physical realistic values. Here you apply to the uncorrected EC observations, which may be physically unrealistic as this is the reason that you want to correct. Please clarify.

11) S2.5: To calculate the energy imbalance (epsilon) the authors also needs the available energy which is built from net radiation and the heat stored into the vegetation-soil system. I miss in section 2 a description of these observation for each site including error characteristics. It would greatly help when for each site a characteristic diurnal cycles are displayed of the components of the energy balance and the resulting imbalance (epsilon). This then should include a discussion if any peculiarities show up in these observations.

12) S2.5: The authors state that they are not interested in analyzing the full energy budget, but only the evaporative component. Alternatively the authors could have chosen to analyze the relation between the lysimeter and EC measurements. It would be nice if the authors could discuss the arguments for choosing not to follow that line.

13) Table 2 – 5: Only very little comments are given by the authors to these eight tables. Some more wording to guide the reader towards important points to learn from each
table would be very helpful.

14) Figure 1a and 2a: The larger differences in the morning in fig 1 have disappeared in figure 2a. This must be related to the diurnal characteristics of epsilon. Addition of epsilon in fig 1 and discussion would be helpful.

15) S3.7 In section 2.5 the authors describe a binning procedure of the LE data for regressing and obtain wL. How does this relate to figure 5 where averages of wL per hour are given. Some extra wording would be helpful.

16) L267: Bins ranging from 6 to 14. Is this the number of observations in each bin. Please be precise.

17) L273: Standard deviation in wL will among others depend on the statistical noise in the EC measurements. These can be large under convective low wind conditions during day time, and lower under the less convective conditions around sun rise and sunset.

18) Figure 6b: what is the meaning of _s11 in the labels?

19) Figure 6b: there is a remarkable drop in wL observed in the figure, but not mentioned in the text.

20) S3.9: Please explain what the value of these correlations are. One question that comes to my mind is: the authors use the result of the regression (wL and cLE) and look at their correlation. What can we learn from this?

21) L315-319: What conclusions can be drawn from the summary of these results on standard deviations?

22) L345: See my comment #17

23) L347: "one might conclude that the high standard variations are rather related to weather conditions". Where is this conclusion based on?
24) S5, L352: I would say that the best adjustment of EC to LY would be a direct regression of without the complications of epsilon and the full energy balance. And if this is the aim, why not use LY and refrain from EC?

25) L358: Note that also the statistics of EC observations will be come progressively worse when going to smaller time intervals. But combining scintillometry and EC-observations might be a way forward.

Textual comments:

26) L14: "At the overall average" -> "Overall" 27) L15: "which were partially closed with" -> "after applying " 28) L16: "remain high differences" -> "remain large differences" 29) L18: "correction evaporation weights". This looks like a defining term, but is never used in the main text, please be concise on terminology. 30) L19: "correcting evaporation weights". Yet another formulation never used in the main text. 31) L29: How is the energy balance gap defined? I would expect a value of 22-27 % for the magnitude of the gap. 32) L30: A comparison alone cannot lead to any reduced difference. I guess it is the adjustment of EC measurements with LY measurements that leads to this reduced difference. 33) L31: How do this percentages relate to the values of 73.2 and 78% on line L29. 34) L35: "with" -> "of" 35) L36: "an influence of the increasing plant height as against constant measurement height is suspected." Unprecise wording, please correct. 36) L38: is -17% to -19% on a daily basis? Please be precise in formulation 37) L65: Textually it would be nicer to start with some of the general information given below the table 1, and then introduce table 1. 38) Table 2b: Some numbers are out of place in the last column, it seems. 39) L281: 212 weeks?

---

## Author Comment (AC3) · 13 Oct 2020

We thank for the careful, profound and challenging review of reviewer #2, which underlines our intentions (1) to draw attention to the subject and (2) demonstrate that the method proposed gives plausible results under various local conditions. We acknowledge their input on the discussion on the standard deviations, which made the manuscript more concise.

Some comments suggest adding additional figures and tables. We are, however,

concerned about the length of the manuscript. Nevertheless, we will do our best to respond to reviewer 2 (italic). In case we cannot fulfill all expectations, we suggest to publish the paper as a technical note. We respond to the referee's comments (italic) below.

Note that the new nomenclature for the variables (e.g. $LE_{LY}$ instead of $LY$, as changed upon input from Reviewer #1) is not yet used here for simplicity.

*Main critics:*

*1) A lot of results are presented in this paper, but often without much comments by the authors. In that respect the paper looks more like a technical report which may form the basis of peer-reviewed paper. I urge the authors to take the reader along the circa eight tables and describe in text what the main message of each table is.*

We changed the text accordingly at several places, see responses below.

*2) Although the authors focus on the relation between LY and EC measurements, they also use the other observations of the surface energy balance, net radiation and surface soil heat flux in essential parts of their analysis. These observations should be described aswell in section 2.*

At the end of section 2.2 we add the information on instrumentation along the lines of: EC fluxes were measured with CSAT-3 ultrasonic anemometers (Campbell Scientific Inc., USA) and LI-7500 infrared gaz analysers (LI-COR Biosciences, USA) at Fendt, Graswang and Rietholzbach. . ...
We are still collecting the necessary details.

*3) A rational for the used method is lacking, given that the authors state that they are mainly interested in the relation of LY and EC evaporation observations.*

We want to demonstrate the applicability of approach described by Widmoser et al (2018) to several stations as mentioned in the abstract. In our opinion, the rationale for this study is made very clear.

*4) Section 4 is more a summary of results then a discussion. For example the part on standard deviations needs a discussion on what these comparison of SDV means and what can be learned of it. Now there are so many nice statistical results and apparently so little conclusions can be drawn. The question is whether these statistical techniques alone are sufficient to grasp the mechanisms behind the differences observed. Perhaps these should be accompanied by detailed case to case studies.*

It is beyond the scope of this article to grasp all mechanisms behind the differences observed. We explain the meaning of SD-differences. See comment 21.

*5) The text is not always as precise at it could be, some examples are given below. But there are more of these occasions. Please copy edit the text carefully on this aspect.*

We will consider all comments below.

*Specific comments*

*6) L45: How is the evaporation fraction used to correct? Is that different from Bowen ratio preservation?*
This comment refers to our citation of Gebler et al (2015). We refer to p. 2151, 3 lines above Eq. 10 in Gebler. Their considerations are rather complex and we prefer not to explain them in detail in our article.

*7) Table 1: It would be nice to have the other information (measurement time interval, vegetation type, period of the day used) also in the table. This may require to turn the table by 90 degrees.*

We incorporate period of day, time interval and vegetation into Table 1 – or we add one table (1b) below Table 1 listing the requested information for all stations.

*8) What would be the influence of the oak trees at station Majadas on the flux observations.*

It is not the scope of this manuscript to investigate these influences and we refer to authors involved in the measurements at the site. However, since the oak trees are rather widely dispersed, we assume a mild influence on turbulence that would be significant to differences between lysimeter and EC measurements.

*9) S2.2: Here general error characteristics are given. Are the authors sure that these can applied to the various sites used here. Are there any specific circumstances which may have an influence on the error characteristics. For example, how well are the conditions in the lysimeter kept comparable to the surrounding fields. Are there infrared surface temperature observations to judge possible inhomogeneities between lysimeter and surroundings?*
We are in contact with the PIs of the stations used in this study and trust that all instruments are maintained with the care needed. This includes keeping the lysimeter vegetation comparable to the surrounding area. Investigating differences between LY and EC originating from such conditions is not the scope of this article. We are citing articles dealing with this problem (Evett et al. 2012) and other sources of uncertainty of lysimeters.

*9) S2.3: Here I have the same questions. I find these error estimates to general. It is always good to look at specifics of datasets/sites. EC measurements require all kinds of corrections. I miss a statement on the applied methods, and any differences in treatment per site.*

We are aware of the extensive correction procedure that is required to obtain accurate flux measurements when using the eddy-covariance method. However, we are in contact with the data providers of every single station presented and trust their capacity of correctly processing theses eddy-covariance data. Describing the flux corrections in detail would take up an unnecessarily large part of the article.

*10) L113: Wohlfahrt and Widmoser (2013) apply the out-of-bound concept for corrected EC observations to judge whether this corrections lead to physical realistic values. Here you apply to the uncorrected EC observations, which may be physically unrealistic as this is the reason that you want to correct. Please clarify.*

Our screening according to the Out-of-Bound-concept avoids data combinations which correspond to case 2 and case 3 in Fig. 1 of Wohlfahrt and Widmoser (2013).

We add at L113: According to this concept, the ratio $r_1 = (r_a + r_c)/r_a$, where $r_a$ and $r_c$ denote aerodynamic and canopy resistance, must numerically be within the range of 1 to infinity (see Fig. 1 in Wohlfahrt and Widmoser, 2013). Case 2 represents $r_1 < 0$ and case 3 represents $0 < r_1 < 1$. Data corresponding to case 2 and 3, are thus omitted.

*11) S2.5: To calculate the energy imbalance (epsilon) the authors also needs the available energy which is built from net radiation and the heat stored into the vegetation-soil system. I miss in section 2 a description of these observation for each site including error characteristics. It would greatly help when for each site a characteristic diurnal cycles are displayed of the components of the energy balance and the resulting imbalance (epsilon). This then should include a discussion if any peculiarities show up in these observations.*

We add figures of the diurnal cycles for observed $A, H, LE, LY$, and epsilon for all four stations in section 2 (see figures at the end).

*12) S2.5: The authors state that they are not interested in analyzing the full energy budget, but only the evaporative component. Alternatively the authors could have chosen to analyze the relation between the lysimeter and EC measurements. It would be nice if the authors could discuss the arguments for choosing not to follow that line.*

Certainly, one can achieve good correlations between $LY$ and $EC$. But omitting epsilon-values would exclude any information about the contribution of $LE$ ($EC$-measurements) to the energy closure. This is, however, the main objective of our paper.

*13) Table 2 – 5: Only very little comments are given by the authors to these eight*

*tables. Some more wording to guide the reader towards important points to learn from each table would be very helpful.*

We add to L173 (Tables 2): They highlight the substantial difference between the humid and dry stations in terms of the mean magnitude of evapotranspiration. Under moist conditions, in contrast, the dry station Majadas (M4) ranges around the same magnitude as the humid stations.We re-formulate L199 (Tables 3): Tables 3a and 3b show the absolute differences and their standard deviation between the $EC$-data presented in Tables 2a and 2b and $LY$-measurements. They indicate how the differences between $EC$- and $LY$-measurements mostly (except for F1) get smaller from observed ($DLoL$) to adjusted values of $LY$ ($DLaL$).

For Tables 4, L228 is moved to L219 (beginning of the corresponding section).

In general, the findings from the tables are presented in the discussion.

*14) Figure 1a and 2a: The larger differences in the morning in fig 1 have disappeared in figure 2a. This must be related to the diurnal characteristics of epsilon. Addition of epsilon in fig 1 and discussion would be helpful.*

The change from the original data ($oLE$) to corrected ($cLE$) and adjusted ($aLE$) data involves the interactions of $oLE, LY, epsilon, d$ and the regression of binned data. Especially the latter smoothes down the morning data.

*15) S3.7 In section 2.5 the authors describe a binning procedure of the LE data for regressing and obtain $wL$. How does this relate to figure 5 where averages of wL per*

*hour are given. Some extra wording would be helpful.*

In line 266 we re-formulated the first sentence to: Figures 5a and 5b show the mean course of $wL$ during daytime-hours using the average of all $wL$ values at a specific hour.

*16) L267: Bins ranging from 6 to 14. Is this the number of observations in each bin. Please be precise.*

This is the numbers of bins. The bins were selected such that the number of data within all bins remained more or less the same. Depending on the sample size we had 90 to ca. 120 observations within a bin. This is described in section 2.5 (see pp. 148-150). For clarity, we re-formulated L267 to: The number of bins used in Fig. 5a per station varies from 6 (F1), 8 (F2, G2, RHB) to 14 (G1). The number of bins used for Majadas in Fig. 5b varies from 5 to 12, depending on the used period.

*17) L273: Standard deviation in wL will among others depend on the statistical noise in the EC measurements. These can be large under convective low wind conditions during day time, and lower under the less convective conditions around sun rise and sunset.*

We are grateful for this hint. However, it's rather during low turbulence conditions (evening through morning), that the signal to noise ratio of the $EC$ system is small and $EC$ measurements inaccurate, since the fluxes are very small. Eddy correlation in low turbulence conditions can then result in flux overestimation. We rather believe that the variability is lower in the morning/evening due to the mentioned smallness of the absolute values during stable/less convective conditions. We add in L275: . . ., which

relates to the fact that the absolute differences between $LY$ and $EC$ observations are comparably small during stable to weakly unstable conditions in the morning and evening. Additionally, in the discussion L343 is re-formulated accordingly, basically dropping the surprising character and deleting the vague statement about weather conditions (see also comments 22 and 23).

*18) Figure 6b: what is the meaning of $_s11$ in the labels?*

$s_11$ stands for data smoothed with a moving median filter with a window length of 11 timesteps. We add this information in the legend: A moving median filter ($s11$) with a window length of 11 hours was used for smoothing.

*19) Figure 6b: there is a remarkable drop in wL observed in the figure, but not mentioned in the text.*

We add a comment on this at L347: The correlation of $wL$ to the magnitude of evaporation is also indicated in Fig. 6b, where a drop in $wL$ follows $cLE$. This correlation is indicated in Fig. 6b, where a drop in $wL$ follows $cLE$.

*20) S3.9: Please explain what the value of these correlations are. One question that comes to my mind is: the authors use the result of the regression (wL and cLE) and look at their correlation. What can we learn from this?*

The authors rather use the regression of $wL$ and $LE$ and then derive $cLE$. The tables reveal, that wL and the magnitude of evaporation are positively correlated. But indeed, these tables present auto-correlated values, and their contribution to the study is small.

We thus drop the whole section 3.9 and mention the correlation $wL$ to magnitude evaporation in the discussion as the statement from comment 19).

*21) L315-319: What conclusions can be drawn from the summary of these results on standard deviations?*

We add the following statement on L319: The difference between $SD(LE)$ and $SD(oLE)$ is getting bigger since $SD(oLE)$ gets smaller after correction, whereas $SD(LE)$ remains the same.

*22) L345: See my comment #17*

See our comment under 17.

*23) L347: "one might conclude that the high standard variations are rather related to weather conditions". Where is this conclusion based on?*

We will drop this statement in view of our new statement on $SD$ under comment 17.

*24) S5, L352: I would say that the best adjustment of EC to LY would be a direct regression of without the complications of epsilon and the full energy balance. And if this is the aim, why not use LY and refrain from EC?*

This is true, but without including epsilon one cannot get the contribution of $EC$ to the energy gap, which is a strong part of the current study. And as the energy gap problem is characteristic to involving $EC$ measurements, using lysimeters for evaporation

instead of EC would obviously solve a part of the problem. It's a good point and we thus add in L364: If a high-precision lysimeter capable of resolving evapotranspiration as well as condensation is available complementary to an $EC$ set-up, $LE$ can directly be obtained from the lysimeter.

*25) L358: Note that also the statistics of EC observations will be come progressively worse when going to smaller time intervals. But combining scintillometry and EC-observations might be a way forward.*

The actual used time interval of $EC$ measurements would not change, only the averaging window on the rawdata could be shifted, as mentioned in L360, so that an $EC$ average would better correspond to a e.g. 5 min lysimeter value. The sentence is unclear and is re-forumalted to: In a first step we recommend to perform the comparison of $LY$ and $EC$ based on 5 to 10 minutes lysimeter intervals, and center the one/half-hourly averaging window on $EC$ raw data accordingly. The problem we see with scintillometry we see in regard of reducing the energy gap is the even larger discrepancy of footprints between $EC$ and scintillometer.

*Textual comments:*

*26) L14: "At the overall average" –> "Overall"*

Good suggestion, we use only 'overall'.

*27) L15: "which were partially closed with" –> "after applying "*

Good suggestion, as the term closed in regard of the measurements is misleading. Chnaged accordingly.

*28) L16: "remain high differences" –> "remain large differences"*

Will be changed.

*29) L18: "correction evaporation weights". This looks like a defining term, but is never used in the main text, please be concise on terminology.*

True, we drop the 'correction'.

*30) L19: "correcting evaporation weights". Yet another formulation never used in the main text.*

True, we drop the 'correcting'.

*31) L29: How is the energy balance gap defined? I would expect a value of 22-27 % for the magnitude of the gap.*

Good catch. This is true and we are aware of this. The numbers used refer to the closure, the gap thus is 22.0 to 27, as the R2 correctly suggests.

*32) L30: A comparison alone cannot lead to any reduced difference. I guess it is the adjustment of EC measurements with LY measurements that leads to this reduced*

*difference.*

This has also been critized by R1 and has been changed accordingly (see also response to R1): After forced closure of the energy balance, the difference between daytime LY and EC data on two fields could be reduced from -28.8% to 6.2%, respectively from -26% to -12.3%, with an accuracy. . ..

*33) L31: How do this percentages relate to the values of 73.2 and 78% on line L29.*

This is now answered with the re-formulation in comment 32).

*34) L35: "with" –> "of"*

Changed.

*35) L36: "an influence of the increasing plant height as against constant measurement height is suspected." Unprecise wording, please correct.*

We re-formulate to: They reported substantially larger $LY$ evapotranspiration rates compared to the EC measurements due to differences in plant growth in the $LY$ and the $EC$ footprint.

*36) L38: is -17% to -19% on a daily basis? Please be precise in formulation*

In L34 we add that Evett used daytime values. Additionally, we re-formulate the

sentence to: …mean differences from -17 to -19 % were found between the two measurements methods after…

*37) L65: Textually it would be nicer to start with some of the general information given below the table 1, and then introduce table 1.*

Good suggestion: We move the first paragraph (In addition to…) to the end of the section, and start the section with the second paragraph (Data were obtained from the following Institutions), foolowed by: Table 1 gives an overview on the location of the sites and time periods used.

*38) Table 2b: Some numbers are out of place in the last column, it seems.*

Those are line numbers being somehow jumbled into the table.

*39) L281: 212 weeks?*

Yes. Is changed to days.
* * *
[Figure]

[Figure]

Y-axis: $W/m^2$ (values: -50, 0, 50, 100, 150, 200, 250, 300, 350, 400, 450, 500)

X-axis: daytime-hours (values: 5, 8, 11, 14, 17, 20)

**Fig. 1.** Average daytime course of available energy oA, sensible heat flux oH, EC-based (oLEEC) and lysimeter-based (LELY) latent heat flux and the energy gap epsilon at Fendt.

**Fig. 2.** Average daytime course of available energy oA, sensible heat flux oH, EC-based (oLEEC) and lysimeter-based (LELY) latent heat flux and the energy gap epsilon at Graswang.

**Fig. 3.** Average daytime course of available energy oA, sensible heat flux oH, EC-based (oLEEC) and lysimeter-based (LELY) latent heat flux and the energy gap epsilon at Majadas.

Fig. 4. Average daytime course of available energy oA, sensible heat flux oH, EC-based (oLEEC) and lysimeter-based (LELY) latent heat flux and the energy gap epsilon at Ri-etholzbach.